# PROBABILISTIC CONFORMAL PREDICTION WITH APPROXIMATE CONDITIONAL VALIDITY

**Vincent Plassier[3]*  Alexander Fishkov[1,4]*  Mohsen Guizani[1]  Maxim Panov[1]  Eric Moulines[2,1]**

[1]Mohamed bin Zayed University of Artificial Intelligence  [2]CMAP, École Polytechnique
[3]Lagrange Mathematics and Computing Research Center
[4]Skolkovo Institute of Science and Technology
maxim.panov@mbzuai.ac.ae    eric.moulines@polytechnique.edu

## ABSTRACT

We develop a new method for generating prediction sets that combines the flexibility of conformal methods with an estimate of the conditional distribution $P_{Y|X}$. Existing methods, such as conformalized quantile regression and probabilistic conformal prediction, usually provide only a marginal coverage guarantee. In contrast, our approach extends these frameworks to achieve approximate conditional coverage, which is crucial for many practical applications. Our prediction sets adapt to the behavior of the predictive distribution, making them effective even under high heteroscedasticity. While exact conditional guarantees are infeasible without assumptions on the underlying data distribution, we derive non-asymptotic bounds that depend on the total variation distance between the conditional distribution and its estimate. Using extensive simulations, we show that our method consistently outperforms existing approaches in terms of conditional coverage, leading to more reliable statistical inference in a variety of applications.

## 1 INTRODUCTION

Conformal prediction methods are widely used to construct prediction sets because they provide finite-sample validity under minimal distributional assumptions (Vovk et al., 2005; Shafer & Vovk, 2008). The split-conformal approach leverages a calibration dataset of size $n$, denoted as $\{(X_k, Y_k)\}_{k \in [n]}$ with $X_k \in \mathbb{R}^d$ and $Y_k \in \mathcal{Y}$, to construct prediction sets $\mathcal{C}_\alpha(x)$ at a specified confidence level $\alpha \in (0, 1)$. Given a conformity score function $V \colon \mathbb{R}^d \times \mathcal{Y} \to \mathbb{R}$, the conformal prediction set for a test point $x \in \mathbb{R}^d$ is defined as:

$$\mathcal{C}_\alpha(x) = \Big\{ y \in \mathcal{Y} \colon V(x, y) \leq Q_{1-\alpha}\Big(\frac{1}{n+1} \sum_{k=1}^n \delta_{V(X_k, Y_k)} + \frac{1}{n+1} \delta_\infty\Big)\Big\}, \tag{1}$$

where $\delta_x$ denotes the Dirac mass at $x$, and $Q_{1-\alpha}(\mu)$ represents the $(1 - \alpha)$-quantile of the probability measure $\mu$. This formulation guarantees valid marginal coverage while being computationally efficient. However, its performance may degrade under distributional heterogeneity, motivating extensions to adapt to varying noise levels. If the calibration data $\{(X_k, Y_k)\}_{k \in [n]}$ is drawn i.i.d. from a population distribution $P_{X,Y}$, then for any new data point $(X_{n+1}, Y_{n+1}) \sim P_{X,Y}$ sampled independently of the calibration data, the conformal theory ensures the *marginal validity* of $\mathcal{C}_\alpha(X_{n+1})$, meaning that $\mathbb{P}(Y_{n+1} \in \mathcal{C}_\alpha(X_{n+1})) \geq 1 - \alpha$. This marginal guarantee can hide significant discrepancies in the coverage of different regions of the input space $\mathbb{R}^d$; see e.g. (Izbicki et al., 2022; Hore & Barber, 2024). Conditional validity is a more desirable guarantee than marginal validity: for any $x \in \mathbb{R}^d$, the set $\mathcal{C}_\alpha(x)$ is *conditionally valid* if

$$\mathbb{P}(Y_{n+1} \in \mathcal{C}_\alpha(X_{n+1}) \mid X_{n+1} = x) \geq 1 - \alpha. \tag{2}$$

However, this property cannot be achieved without further assumptions about the data distribution; see (Vovk, 2012; Lei & Wasserman, 2014). For practical purposes, it is enough to construct sets $\mathcal{C}_\alpha$ which approximate (2), and ideally achieve it asymptotically under suitable conditions in the limit of large sample size $n$.

---

*Equal contribution

RELATED WORK

**Conformal prediction with conditional guarantees.** Much research has been devoted to this problem, starting with the case where $\mathcal{Y} = \mathbb{R}$; see (Foygel Barber et al., 2021). For example, Romano et al. (2019) and Kivaranovic et al. (2020) proposed methods based on estimates of the lower and upper conditional quantile functions which are then conformalized. Sesia & Candès (2020) have shown that the constructed interval converges to the narrowest possible interval that achieve conditional coverage. We stress that these methods are specific to the case $\mathcal{Y} = \mathbb{R}$. In addition, when the conditional distribution $P_{Y|X}$ is multimodal, restricting prediction to intervals is suboptimal; see (Wang et al., 2023) for a discussion.

**Conformal prediction based on estimates of conditional distribution.** A number of studies have focused on the construction of prediction sets using an estimator of the conditional distribution $P_{Y|X}$. In the case where $\mathcal{Y} = \mathbb{R}$, Cai et al. (2014) and Lei & Wasserman (2014) have constructed prediction intervals based on a conditional density estimator and have established their asymptotic validity under appropriate conditions. More recently, Han et al. (2022) have employed kernel density estimation to construct asymmetric prediction bands. However, this method is inherently limited as it produces a single interval and thus cannot effectively capture the multimodality of the predictive distributions. On the other hand, Sesia & Romano (2021) partition the domain of $Y$ into bins, forming a histogram-based approximation of $P_{Y|X}$. The authors demonstrated that their method satisfies marginal validity while achieving asymptotic conditional coverage; see also (Lei et al., 2018). Asymptotic conditional coverage is also obtained using cumulative distribution function estimators (Izbicki et al., 2020; Chernozhukov et al., 2021). Guha et al. (2024) introduced an approach that reformulates regression tasks as classification problems by discretizing the output space into bins. This discretization enables the approximation of the conditional density, facilitating the construction of prediction sets that align with the highest posterior density (HPD) regions. However, a key limitation of this method is the computational cost associated with fine-grained discretization, as the number of labels required for complex scenarios can become prohibitive. In a complementary direction, Kiyani et al. (2024) proposed a method to enhance adaptive coverage by learning a family of weights that dynamically adjust the quantile of the conformity score based on the covariate $x$. Their approach ensures that the conditional coverage remains close to the target level $1 - \alpha$. (Guan, 2023; Alaa et al., 2023; Hore & Barber, 2024) develop a localized conformal inference approach that partitions feature space and applies conformal prediction within neighborhoods, aiming to achieve closer-to-conditional validity by accounting for conditional distribution; see (Romano et al., 2020a; Melki et al., 2023).

**Conformal prediction for multi-output regression.** Very few studies have explored the setting where the prediction target is multi-dimensional, i.e., $\mathcal{Y} = \mathbb{R}^q$ with $q > 1$. Wang et al. (2023) proposed the PCP method, which leverages implicit conditional generative models (CGMs) to generate samples from the conditional distribution. The method constructs prediction sets as unions of balls centered at CGM-generated samples. However, a key limitation of PCP lies in using a fixed-radius ball across the space. This design can lead to over-coverage in low-variability regions, where a smaller radius would suffice, and under-coverage in highly dispersed areas, where a larger radius is needed to capture the conditional distribution fully. This underscores the need for a more adaptive methodology to adjust the size of prediction sets in response to local variability in the data.

Our work addresses these challenges through the following main **contributions**.

- We propose a new $CP^2$ method for constructing confidence sets that adapt to the local structure of the data distribution, capable of addressing complex multi-dimensional prediction tasks where $\mathcal{Y} = \mathbb{R}^q$. Our approach is versatile in accommodating scenarios involving either an explicit conditional density estimator or an implicit generative model; see Section 2.

- We develop a theoretical framework to analyze the properties of the proposed $CP^2$ method, establishing both its marginal and approximate conditional validity. Furthermore, we demonstrate that asymptotic conditional coverage is attainable under a weak consistency assumption on the predictive distribution; see Section 3.

- We demonstrate the effectiveness of the proposed method through a series of experiments on synthetic and real-world datasets. The results indicate that our approach consistently outperforms existing methods in terms of conditional coverage. Specifically, it excels in handling classical regression problems, effectively addressing multimodality, and proves robust in the more challenging setting of multidimensional prediction tasks; see Section 4.

## 2 THE $\mathrm{CP}^2$ FRAMEWORK

### 2.1 CONFORMAL PREDICTION ASSISTED WITH DISTRIBUTION ESTIMATOR

**Problem setup.** Our goal is to construct marginally valid predictive sets with approximate conditional validity. To achieve this, we adopt the split-conformal approach to conformal inference (Papadopoulos et al., 2002; Papadopoulos, 2008; Romano et al., 2019; Plassier et al., 2024). Specifically, we partition the data into two disjoint subsets: a training set, $\mathcal{T} = \{(\tilde{X}_k, \tilde{Y}_k)\}_{k=1}^m$, and a calibration set, $\mathcal{C} = \{(X_k, Y_k)\}_{k=1}^n$. We assume that the training and calibration samples are mutually independent and i.i.d. according to the distribution $P_{X,Y}$ over the feature space $\mathbb{R}^d$ and response space $\mathcal{Y}$. The target space $\mathcal{Y}$ may be either discrete or continuous, allowing for broad applicability of the method.

**Conditional distribution estimator.** We assume that an estimator $\Pi_{Y|X}$ of the conditional probability $P_{Y|X}$ is learned from the training data $\mathcal{T}$. A well-established body of research exists on nonparametric conditional density estimation, with classical approaches relying on smoothing techniques such as kernel smoothing and local polynomial fitting; see e.g. (Rosenblatt, 1969; Hyndman et al., 1996; Fan & Yim, 2004). An alternative strategy reformulates conditional density estimation as a regression task, enabling the use of nonparametric regression methods to approximate the conditional density. More recently, generative approaches based on deep neural networks have been proposed, allowing for efficient sampling from the conditional distribution; see (Abadi et al., 2016; Zhou et al., 2021) for examples. In what follows, we treat the choice of this estimator / generator as a black box.

**Motivation: Probabilistic Conformal Prediction (`PCP`).** Having access to samples from the conditional distribution, one, following Wang et al. (2023), may define the prediction set as $\mathcal{R}_Z(x; t) := \cup_{i=1}^M \mathrm{B}(\hat{Y}_i, t)$, where $\mathrm{B}(y, t)$ is a ball of radius $t$ in $\mathcal{Y}$ centered at $y$. The set $\mathcal{R}_Z(x; t)$ and it depends on exogenous variables $Z = (\hat{Y}_1, \ldots, \hat{Y}_M)$, where each $\hat{Y}_i$ is sampled conditionally independently from the conditional generative model $\Pi_{Y|X=x}$. Now one can define a confidence score $V_Z(x, y) = \inf\{t \geq 0 : y \in \mathcal{R}_Z(x; t)\}$. In the case of `PCP` the score becomes $V_Z(x, y) = \min_{i=1}^M \|y - \hat{Y}_i\|$. The conformity scores $V_{Z_k}(X_k, Y_k)$ can be used in split-CP framework straight ahead. The resulting confidence sets $\mathcal{C}_\alpha(x)$ given by (1) satisfy marginal coverage guarantees if marginalization is done over all the random variables involved, i.e. $X, Y$ and $Z$. This confidence set adapts to the conditional distribution at a given point as it depends on samples from this distribution. However, still the sets at different points share the same radius parameter $t$. The further adaptation of pointwise radius can improve conditional coverage.

**Towards conditional coverage.** Conditional coverage is automatically satisfied for methods based on confidence scores of the form $V_Z(x, y) = \inf\{t \geq 0 \mid y \in \mathcal{R}_Z(x; t)\}$, provided that one considers the oracle set:

$$\mathcal{C}_\alpha(x) = \{y \in \mathcal{R}(x; t^*_{x,Z})\}, \tag{3}$$

where $t^*_{x,Z}$ is defined using the true predictive distribution $P_{Y|X=x}$ as:

$$t^*_{x,Z} = \inf\{t \geq 0 \mid P_{Y|X=x}(\mathcal{R}_Z(x; t)) \geq 1 - \alpha\}. \tag{4}$$

The oracle set (3) guarantees both conditional and marginal validity. However, in practical settings, the true conditional distribution $P_{Y|X=x}$ is unknown, and only an estimate $\Pi_{Y|X=x}$ is available. Substituting $\Pi_{Y|X=x}$ directly into (4) and (3) introduces estimation errors, which can compromise both marginal and conditional coverage guarantees.

### 2.2 $\mathrm{CP}^2$ FRAMEWORK

There are several main ingredients for our approach:

1. In classical conformal prediction, the shape of the prediction set is determined by the score function $V(x, y)$; see (1). Our approach builds on this framework by introducing a family of confidence sets $\mathcal{R}_z(x; t)$, explicitly parameterized by $t \in \mathsf{T}$, where $\mathsf{T} \subseteq \mathbb{R}$ and $z \in \mathcal{Z}$ represents a vector of auxiliary variables. In most cases, the index set can be chosen as either $\mathsf{T} = \mathbb{R}$ or $\mathsf{T} = \mathbb{R}_+$. For these confidence sets to be well-defined and meaningful, we impose the following key assumptions: (a) *Monotonicity*: The size of $\mathcal{R}_z(x; t)$ increases with $t$ for any $z \in \mathcal{Z}$. Furthermore, the entire output space $\mathcal{Y}$ is covered by $\cup_{t \in \mathsf{T}} \mathcal{R}_z(x; t)$; (b) *Continuity*: The mapping $t \mapsto \mathcal{R}_z(x; t)$ exhibits a suitable form of continuity. Specifically, we assume the following mathematical property:

Table 1: Confidence sets $\mathcal{R}(x;t)$ found in the literature and also discussed in (Gupta et al., 2022).

| Lei et al. (2018) | Lei et al. (2018) | Kivaranovic et al. (2020) |
|---|---|---|
| $[\mathrm{pred}(x) - t, \mathrm{pred}(x) + t]$ | $[\mathrm{pred}(x) - t\sigma(x), \mathrm{pred}(x) + t\sigma(x)]$ | $(1+t)[q_{\alpha/2}(x), q_{1-\alpha/2}(x)] - tq_{1/2}(x)$ |
| Chernozhukov et al. (2021) | Romano et al. (2019) | Sesia & Candès (2020) |
| $[q_t(x), q_{1-t}(x)]$ | $[q_{\alpha/2}(x) - t, q_{1-\alpha/2}(x) + t]$ | $[q_{\alpha/2}(x), q_{1-\alpha/2}(x)] \pm t(q_{1-\alpha/2}(x) - q_{\alpha/2}(x))$ |

**H 1.** For any $(x, z) \in \mathbb{R}^d \times \mathcal{Z}$, the confidence sets $\{\mathcal{R}_z(x;t)\}_{t \in \mathsf{T}}$ are non-decreasing, $\Pi_{Y|X=x}(\cap_{t \in \mathsf{T}} \mathcal{R}_z(x;t)) = 0$, $\cup_{t \in \mathsf{T}} \mathcal{R}_z(x;t) = \mathcal{Y}$. In addition, for any $t \in \mathsf{T}$, $\cap_{t' > t} \mathcal{R}_z(x;t') = \mathcal{R}_z(x;t)$.

*Example* 2.1. For instance, $\mathcal{R}(x;t)$ can be chosen as a ball centered around an estimate of conditional mean $\mathrm{P}_{Y|X}$ with radius $t$. There is no auxiliary variables then an we remove subscript $z$. Examples of confidence intervals specialized to the case where $\mathcal{Y} = \mathbb{R}$ are given in Table 1.

*Example* 2.2. If the predictive distribution is multimodal, a ball centered around the predictive mean often fails to provide an informative prediction set. Ideally, $\mathcal{R}_z(x;t)$ should correspond to the set with the highest predictive density (HPD) of $\mathrm{P}_{Y|X}$. However, HPD regions are difficult to determine in practice, even when $\mathrm{P}_{Y|X}$ is available. One of the viable options is given by PCP approach (Wang et al., 2023) for which prediction set is $\mathcal{R}_Z(x;t) := \cup_{i=1}^{M} \mathrm{B}(\hat{Y}_i, t)$ with $Z = (\hat{Y}_1, \ldots, \hat{Y}_M) \in \mathcal{Z}$ and each $\hat{Y}_i$ being an independent sample from $\Pi_{Y|X=x}$.

2. As discussed above, a natural choice for the confidence score is the minimal size of the set required to cover the observation $y$ at the input $x$ for the auxiliary variables $z$: $V_z(x, y) = \inf \{t \in \mathsf{T} \colon y \in \mathcal{R}_z(x;t)\}$.

3. The key step within the proposed approach is to adapt conformal prediction method by pointwise adaptation of the confidence set radius $t$. While the oracle radius (4) is not achievable, we will develop a data-driven procedure to approximate it. First, it is convenient to introduce a function $f_\tau(v)$ parameterized by $\tau$. Such function aims to transform the conformity score $v$ and was introduced (albeit in a slightly different form) in (Han et al., 2022; Deutschmann et al., 2024; Plassier et al., 2025). Examples of such a function are $f_\tau(v) = \tau v$ and $f_\tau(v) = \tau + v$. We assume that:

**H2.** There exists $\varphi \in \mathsf{T}$ such that $\tau \in \mathsf{T} \mapsto f_\tau(\varphi)$ is increasing and bijective. In addition, $v \in \mathsf{T} \mapsto f_\tau(v)$ is increasing for any $\tau \in \mathsf{T}$.

We define $\tau_{x,z}$ using the estimated predictive density $\Pi_{Y|X=x}$ according to

$$\tau_{x,z} = \inf \left\{\tau \in \mathsf{T} \colon \Pi_{Y|X=x}(\mathcal{R}_z(x; f_\tau(\varphi))) \geq 1 - \alpha\right\}. \tag{5}$$

It is easily shown that for any $\alpha \in (0, 1)$, $x \in \mathbb{R}^d$, $z \in \mathcal{Z}$, it holds $\tau_{x,z} \in \mathsf{T}$ and $\Pi_{Y|X=x}(\mathcal{R}_z(x; f_{\tau_{x,z}}(\varphi))) \geq 1 - \alpha$; see Lemma A.3.

4. The resulting procedure works as follows. Let $\bar{\Pi}_{Z|X=x}$ define a probability distribution on $\mathcal{Z}$. The standard choice is to use conditional distribution as a sampler: $\bar{\Pi}_{Z|X=x} = \Pi_{Y|X=x}^{\otimes M}$. For $k \in \{1, \ldots, n\}$, we set $\tau_k := \tau_{X_k, Z_k}$ and $V_k := V_{Z_k}(X_k, Y_k)$, where $\{Z_k\}_{k=1}^{n}$ are sampled conditionally independently from $\bar{\Pi}_{Z|X=X_k}$. Given $X_{n+1} \in \mathbb{R}^d$, we sample $Z_{n+1} \sim \bar{\Pi}_{Z|X=X_{n+1}}$ independently from $\{(X_k, Y_k, Z_k)\}_{k=1}^{n}$, and construct the resulting $\mathrm{CP}^2$ prediction set as

$$\mathcal{C}_\alpha(X_{n+1}) = \mathcal{R}_{Z_{n+1}}\left(X_{n+1}; f_{\tau_{n+1}}\left(Q_{1-\alpha}(\mu_n)\right)\right), \tag{6}$$

where $Q_{1-\alpha}(\mu_n)$ is the $(1 - \alpha)$-quantile of the distribution $\mu_n$ given by

$$\mu_n = \frac{1}{n+1} \sum_{k=1}^{n} \delta_{f_{\tau_k}^{-1}(V_k)} + \frac{1}{n+1} \delta_\infty. \tag{7}$$

The transformation $\{v \mapsto f_\tau(v)\}_{\tau \in \mathsf{T}}$ balances the following two factors: (a) The optimal parameter $V_z(x, y)$ ensuring that $y$ is included in the confidence set $\mathcal{R}_z(x; V_z(x, y))$; (b) The parameter $\tau_{x,z}$ obtained from the probabilistic model $\Pi_{Y|X=x}$.

We stress that $\mathrm{CP}^2$ is a general framework that can be adapted to many choices for conditional predictive density estimates, constructing the family of confidence sets, and selecting the calibration function $f_\tau(v)$. However, we start with the simple example that shows that $\mathrm{CP}^2$ is more general than the classical split-conformal CP approach.

---

**Algorithm 1** $\mathrm{CP}^2\text{-PCP}$

---

**Input:** dataset $\{(X_k, Y_k)\}_{k\in[n]}$, significance level $\alpha$, conditional distribution $\Pi_{Y|X}$, function $f_t$.

// **Compute the** $(1-\alpha)$**-quantile**

**for** $k = 1$ **to** $n$ **do**

    Sample $\{\hat{Y}_{k,i}\}_{i=1}^M$ and $\{\tilde{Y}_{k,j}\}_{j=1}^{\tilde{M}}$ from $\Pi_{Y|X=X_k}$

    Set $V_k = \min_{i=1}^M \|Y_k - \hat{Y}_{k,i}\|$

    Set $\tau_k = (t \mapsto f_t(\varphi))^{-1}\{Q_{1-\alpha}(\tilde{M}^{-1}\sum_{j=1}^{\tilde{M}}\delta_{\min_{i=1}^M\|\tilde{Y}_{k,j}-\hat{Y}_{k,i}\|})\}$

$Q_{1-\alpha}(\mu_n) \leftarrow \lceil(1-\alpha)(n+1)\rceil$-th smallest value in $\{f_{\tau_k}^{-1}(V_k)\}_{k\in[n]} \cup \{\infty\}$

// **Compute the prediction set for a new point** $x \in \mathbb{R}^d$

Sample $z = \{\hat{Y}_i\}_{i=1}^M$ and $\{\tilde{Y}_j\}_{j=1}^{\tilde{M}}$ from $\Pi_{Y|X=x}$

Set $\tau_{x,z} = (t \mapsto f_t(\varphi))^{-1}\{Q_{1-\alpha}(\tilde{M}^{-1}\sum_{j=1}^{\tilde{M}}\delta_{\min_{i=1}^M\|\tilde{Y}_j-\hat{Y}_i\|})\}$

**Output:** $\mathcal{C}_\alpha(x) = \cup_{i=1}^M \mathrm{B}(\hat{Y}_i, f_{\tau_{x,z}}(Q_{1-\alpha}(\mu_n)))$.

---

**Simple example of** $\mathrm{CP}^2$**.** We begin with a simple application of $\mathrm{CP}^2$ to highlight its differences from the basic conformal approach, with $\mathcal{Y} = \mathbb{R}$. The calibration sets are defined as: $\mathcal{R}(x; t) = \{y \in \mathcal{Y}: |y - \mathrm{pred}(x)| \leq t\}$ for $t \in \mathbb{R}_+$. There are no auxiliary variables $z$ in this case, so we omit $z$ from the notation. Assumption **H**1 is easily satisfied with $\mathsf{T} = \mathbb{R}_+$. We then take $f_\tau(v) = \tau v$, $\tau \in \mathbb{R}_+$ and $\varphi = 1$: **H**2 is also satisfied. Note that $f_\tau^{-1}(v) = v/\tau$ for $\tau \in \mathbb{R}_+^*$. Using the $\mathrm{CP}^2$ approach, we find that $V(x, y) = |y - \mathrm{pred}(x)|$, which corresponds to a standard conformity score. The classical conformal prediction method defines the $(1-\alpha)$-quantile based on the associated empirical measure $\nu_n = \frac{1}{n+1}\sum_{k=1}^n \delta_{V_k} + \frac{1}{n+1}\delta_\infty$,

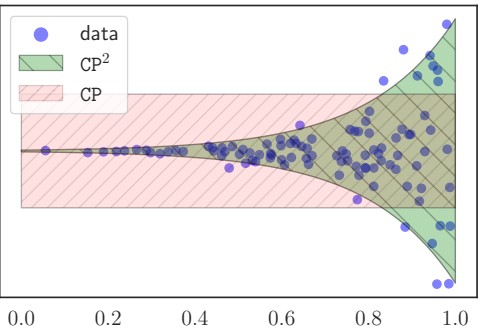

Figure 1: Predictions sets obtained via the standard $\mathrm{CP}$ and $\mathrm{CP}^2$ methods.

where $V_k = |Y_k - \mathrm{pred}(X_k)|$. $\mathrm{CP}^2$ differs from the basic conformal approach by introducing $\tau_x = \arg\min\{\tau \in \mathsf{T}: \Pi_{Y|X=x}([\mathrm{pred}(x) \pm \tau]) \geq 1 - \alpha\}$ as in (5), where $[\mathrm{pred}(x) \pm \tau] = [\mathrm{pred}(x) - \tau, \mathrm{pred}(x) + \tau]$. The prediction set becomes $[\mathrm{pred}(x) \pm f_{\tau_x}(Q_{1-\alpha}(\mu_n))]$, where $\mu_n = \frac{1}{n+1}\sum_{k=1}^n \delta_{V_k/\tau_k} + \frac{1}{n+1}\delta_\infty$. To illustrate the advantage of our method, in Figure 1 we present the prediction sets obtained with the classical CP method and $\mathrm{CP}^2$ in the case of a Neal's funnel-shaped distribution in 2 dimensions; see (Neal, 2003, Section 9).

**$\mathrm{CP}^2$ with Highest Predictive Density regions: $\mathrm{CP}^2\text{-HPD}$.** The natural approach for the general case is to use the conditional distribution $P_{Y|X=x}$ or its estimate $\Pi_{Y|X=x}$ to find Highest Predictive Density (HPD) regions and calibrate their size with the help of $\mathrm{CP}^2$. We develop the respective general algorithm $\mathrm{CP}^2\text{-HPD}$ in Appendix C.1. However, the procedures to find HPDs are usually highly non-trivial and we only investigate this approach experimentally for synthetic data; see Section 4.1. Next, we provide a specific implementation of our general $\mathrm{CP}^2$ framework that is universally applicable.

**$\mathrm{CP}^2$ with Implicit Conditional Generative model: $\mathrm{CP}^2\text{-PCP}$.** We also develop a second instance of the $\mathrm{CP}^2$ algorithm, inspired by Wang et al. (2023). Unlike $\mathrm{CP}^2\text{-HPD}$, this approach does not require the conditional density. It is designed for cases where the conditional generative model (CGM) $\Pi_{Y|X=x}$ is implicit: we cannot evaluate it pointwise while being able to sample from it. For each calibration point $X_k$, we draw $M$ random variables $\{\hat{Y}_{k,i}\}_{i=1}^M$ from $\Pi_{Y|X=X_k}$. We denote $Z_k = (\hat{Y}_{k,1}, \ldots, \hat{Y}_{k,M})$ and consider the confidence sets as the union of balls centered around the sample points $\mathcal{R}_{Z_k}(X_k; t) = \cup_{i=1}^M \mathrm{B}(\hat{Y}_{k,i}, t)$. With such choice, we get $V_k = \min_{i=1}^M \|Y_k - \hat{Y}_{k,i}\|$. We then draw a second sample $\{\tilde{Y}_{k,j}\}_{j=1}^{\tilde{M}}$, and compute $\tau_k = \inf\{t \in \mathbb{R}_+: \tilde{M}^{-1}\sum_{j=1}^{\tilde{M}}\mathbb{1}\{\tilde{Y}_{k,j} \in \mathcal{R}_{Z_k}(X_k; f_t(\varphi))\} \geq 1 - \alpha\}$. It is easily seen that

$$\tau_k = (t \mapsto f_t(\varphi))^{-1}\left\{Q_{1-\alpha}\left(\tfrac{1}{\tilde{M}}\sum_{j=1}^{\tilde{M}}\delta_{\min_{i=1}^M\|\tilde{Y}_{k,j}-\hat{Y}_{k,i}\|}\right)\right\}.$$

Given a new input $X_{n+1} \in \mathbb{R}^d$, we sample $Z_{n+1} = (\hat{Y}_{n+1,1}, \dots, \hat{Y}_{n+1,M})$ and obtain prediction set

$$\mathcal{C}_\alpha(X_{n+1}) = \left\{ y \in \mathcal{Y} \colon \min_{i=1}^M \|y - \hat{Y}_{n+1,i}\| \leq f_{\tau_{n+1}}(Q_{1-\alpha}(\mu_n)) \right\},$$

where $\mu_n$ is given in (7). The $\mathtt{CP}^2\mathtt{-PCP}$ method employs the same type of a confidence set $\mathcal{R}_z(x;t)$ as the one used by $\mathtt{PCP}$ and corresponding confidence score $V_z(x,y)$. However, the key distinction between the two algorithms lies in the additional parameter $\tau_{x,z}$ for $\mathtt{CP}^2\mathtt{-PCP}$, which requires the generation of a second random sample from $\Pi_{Y|X=x}$. This method is especially useful when solving equation (5) is intractable. We summarize $\mathtt{CP}^2\mathtt{-PCP}$ in Algorithm 1.

## 3 THEORETICAL GUARANTEES

In this section, we provide both marginal and conditional guarantees for the prediction set $\mathcal{C}_\alpha(x)$ given in (6). The validity of these guarantees is ensured by the exchangeability of the calibration data, with the exception of Theorem 3.3 which relies on a concentration inequality and thus requires i.i.d. calibration data.

**Marginal and conditional validity of $\mathtt{CP}^2$.** The following theorem establishes *marginal validity* of the predictive set defined by $\mathtt{CP}^2$.

**Theorem 3.1.** *Assume **H1**-**H2**. Then, for any $\alpha \in (0,1)$, it holds $1 - \alpha \leq \mathbb{P}(Y_{n+1} \in \mathcal{C}_\alpha(X_{n+1}))$. Moreover, if the conformity scores $\{f_{\tau_k}^{-1}(V_k)\}_{k=1}^{n+1}$ are almost surely distinct, then it also holds that $\mathbb{P}(Y_{n+1} \in \mathcal{C}_\alpha(X_{n+1})) < 1 - \alpha + (n+1)^{-1}$.*

This is the standard conformal prediction result applied to the scores $\{f_{\tau_k}^{-1}(V_k)\}_{k=1}^{n+1}$ and its proof is postponed to Appendix A.1. Importantly, the upper bound on the coverage always holds when the distribution of $f_{\tau_k}^{-1}(V_k)$ is continuous.

Now, we will investigate the *conditional validity*. Denote by $\mathrm{d}_{\mathrm{TV}}$ the total variation distance and by $\mathbb{P}^{\mathcal{T}}$ the conditional probability given the training data.

**Theorem 3.2.** *Assume **H1**-**H2**, and let $\alpha \in (0,1)$. For any $x \in \mathbb{R}^d$ and $z \in \mathcal{Z}$, it holds*

$$\mathbb{P}^{\mathcal{T}}(Y_{n+1} \in \mathcal{C}_\alpha(x) \mid (X_{n+1}, Z_{n+1}) = (x,z)) \geq 1 - \alpha - \mathrm{d}_{\mathrm{TV}}(\mathrm{P}_{Y|X=x}; \Pi_{Y|X=x}) - p_{n+1}(x,z),$$

*where $p_{n+1}(x,z) = \mathbb{P}^{\mathcal{T}}\left(Q_{1-\alpha}(\mu_n) < f_{\tau_{n+1}}^{-1}(V_{n+1}) \leq \varphi \mid (X_{n+1}, Z_{n+1}) = (x,z)\right)$.*

The proof is postponed to Appendix A.1. This result shows the role of the accuracy of conditional distribution estimator: the more accurately the estimator $\Pi_{Y|X=x}$ approximates the true conditional distribution $\mathrm{P}_{Y|X=x}$, the closer the result will be to $1 - \alpha$. The second term in the lower bound is $p_{n+1}(x,z)$. Its expected value is upper bounded by $\mathbb{E}[p_{n+1}(X,Z)] \leq \alpha$, and non-asymptotic bounds for this error term are developed in Appendix A.2.

**Asymptotic conditional coverage for $\mathtt{CP}^2$.** In the following theorem, we examine the asymptotic conditional conformal validity as the size of the training dataset, $m_n$, goes to infinity with $n$. To make the dependency of the estimator on the size of the training set explicit, we will denote the conditional distribution as $\Pi_{Y|X}^{(m_n)}$. Consider the following assumption.

**H3.** There exists sequence $(r_n)$ such that $\lim_{n \to \infty} \mathbb{P}(\mathrm{d}_{\mathrm{TV}}(\mathrm{P}_{X,Y}; \mathrm{P}_X \times \Pi_{Y|X}^{(m_n)}) \leq r_n) = 1$.

In most interesting case, we have $\lim_{n \to \infty} r_n = 0$. Such types of bounds can be deduced from (Devroye & Lugosi, 2001, Chapter 9). Let $(X,Y,Z)$ and $(X, \hat{Y}, Z)$ be random variables distributed according to $\mathrm{P}_{X,Y} \times \bar{\Pi}_{Z|X}$ and $\mathrm{P}_X \times \Pi_{Y|X}^{(m_n)} \times \bar{\Pi}_{Z|X}$, respectively.

**Theorem 3.3.** *Assume **H1**-**H2**-**H3** hold. If the distributions of $f_{\tau_{X,Z}}^{-1}(V_Z(X,Y))$ and $f_{\tau_{X,Z}}^{-1}(V_Z(X, \hat{Y}))$ are continuous, then, it holds*

$$\left| \mathbb{P}^{\mathcal{T}}(Y_{n+1} \in \mathcal{C}_\alpha(X_{n+1}) \mid X_{n+1}, Z_{n+1}) - 1 + \alpha \right| = \mathrm{O}_{\mathbb{P}}\left(\sqrt{n^{-1} \log n} + r_n\right).$$

In (Lei et al., 2018; Izbicki et al., 2020; Sesia & Candès, 2020), the asymptotic conditional validity is demonstrated by assuming the consistency of their methods' estimators. For instance, Romano et al. (2019) assume that the conditional quantile regressor converges in $L^2$ towards the true quantile with high probability.

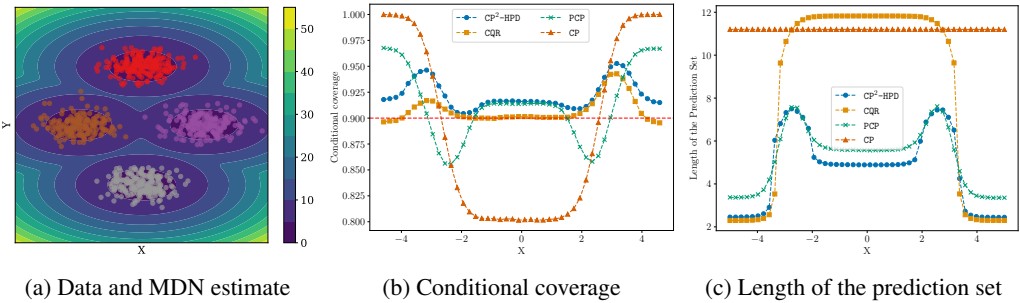

(a) Data and MDN estimate      (b) Conditional coverage      (c) Length of the prediction set

Figure 2: Experimental results on synthetic data in the multimodal case.

## 4 NUMERICAL EXPERIMENTS

In this section, we conduct a comprehensive analysis demonstrating the advantage of $CP^2$ compared to standard and adaptive split conformal algorithms. Specifically, we benchmark our algorithm against several state-of-the-art methods: Conformalized Quantile Regression (Romano et al., 2019), Conformalized Histogram Regression (Sesia & Romano, 2021), Probabilistic Conformal Prediction (Wang et al., 2023) and several others. All these methods share some key aspects: they are built on top of the pre-trained models and do not require access to training data or the model's internals on both calibration and prediction steps. We aim to answer these specific questions: how does $CP^2$ performs in terms of coverage, conditional coverage and predictive set volume when compared to state-of-the-art methods on synthetic and real data[1].

### 4.1 SYNTHETIC DATA EXPERIMENT

In this example, $(X_k, Y_k)$ is sampled from a mixture of $P = 4$ Gaussians; see Figure 2a. The number of training and calibration samples is $m = 10^4$ and $n = 10^3$, respectively. We fit a Mixture Density Network (MDN) as an explicit generative model, $\Pi_{Y|X=x}(y) = \sum_{\ell=1}^{P} \pi_\ell(x)\mathcal{N}(y; \mu_\ell(x), \sigma_\ell^2(x))$, where $\mu_\ell(\cdot)$, $\sigma_\ell(\cdot)$ and $\pi_\ell(\cdot)$ are all modeled by fully connected 2-layers neural networks (the condition $\sum_{\ell=1}^{P} \pi_\ell(x) = 1$ is ensured by using softmax activation functions). We use $CP^2$-HPD (the calculation of the HPD rates as well as $\tau_x$ and $V(x, y)$ is explicit in this case) with $f_t(v) = tv$. The parameters of the MDN are trained by maximizing the likelihood on the training set.

We compare the plain $CP^2$-HPD, PCP (with the same MDN as $CP^2$-HPD and $M = 50$ draws) and CQR. All methods achieve the desired marginal coverage $1 - \alpha = 0.9$. We illustrate the conditional coverage in Figure 2b and the lengths of the predictive sets in Figure 2c. CP with a fixed-width predictive set performs poorly in this multimodal example, both in terms of the size of the confidence set and the conditional coverage $CP^2$-HPD and CQR perform similarly in terms of conditional coverage (which remains close to $1 - \alpha = 0.9$). The conditional coverage of PCP varies between 0.85 and 0.95. $CP^2$-HPD produces smaller prediction sets compared to CQR and PCP as HPD confidence set is more suitable for multimodal applications than the interval produced by CQR.

### 4.2 REAL-WORLD REGRESSION DATA EXPERIMENTS

**Datasets.** We use publicly available regression datasets, which are also considered in (Romano et al., 2019; Wang et al., 2023). Some of them come from the UCI repository: bike sharing (`bike`), protein structure (`bio`), blog feedback (`blog`), Facebook comments (`fb1` and `fb2`). Other datasets come from US Department of Health surveys (`meps19`, `meps20` and `meps21`), and from weather forecasts (`temp`; Cho et al. (2020)).

**Methods.** We compare the proposed $CP^2$-PCP method with Probabilistic Conformal Prediction (PCP; Wang et al. (2023)), Conformalized Quantile Regression (CQR; Romano et al. (2019)), Conformalized Histogram Regression (CHR; Sesia & Romano (2021)), Conformal Prediction with Conditional Guaranties (CPCG; Gibbs et al. (2023)), Localized Conformal Prediction (LCR; Guan

---

[1]Code of experiments can be found at `https://github.com/stat-ml/conditional_cp`

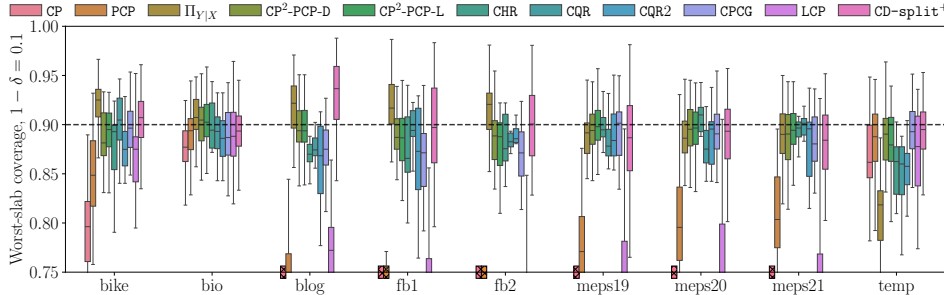

Figure 3: Worst-slab coverage on real data. Results averaged over 50 random splits of each dataset. Calibration and test set sizes set to 2000, 50 conditional samples for $\texttt{PCP}$, $\texttt{CP}^2$ and $\Pi_{Y|X}$. Worst-slab coverage parameter $(1 - \delta) = 0.1$. Nominal coverage level is $(1 - \alpha) = 0.9$ and is shown in dashed black. Methods with conditional coverage below $0.75$ shown as cross-hatched on horizontal axis.

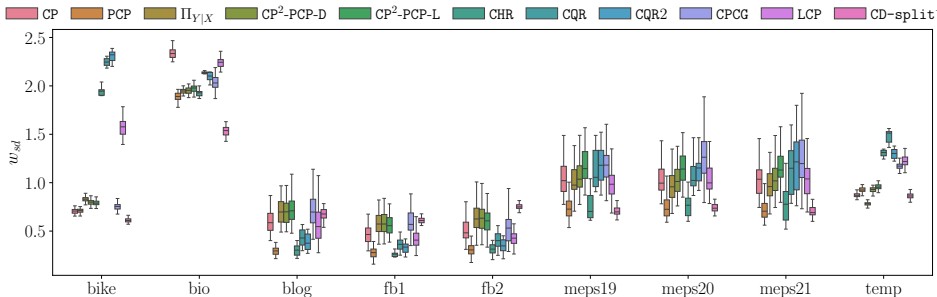

Figure 4: Sizes of the prediction sets on real data. We divide the size of the set by the standard deviation of response to present the results on the same scale.

(2023)) and $\texttt{CDSplit}^+$ (Izbicki et al. (2022)). We also consider $\texttt{CQR2}$ which is a modification of $\texttt{CQR}$ that uses inverse quantile as conformity score. For our method and $\texttt{PCP}$ we use a Mixture Density Network (Bishop, 1994) to estimate the conditional distribution $P_{Y|X}$ that was best-performing in (Wang et al., 2023). We also consider different choices of $f_t$ for our method: $\texttt{CP}^2\texttt{-PCP-L}$ stands for $\texttt{CP}^2\texttt{-PCP}$ with $f_t(v) = tv$ and $\texttt{CP}^2\texttt{-PCP-D}$ stands for $\texttt{CP}^2\texttt{-PCP}$ with $f_t(v) = t + v$. Additionally, we consider $\Pi_{Y|X}$ which is a special case of $\texttt{CP}^2\texttt{-PCP}$ with $f_t(v) = t$.

**Metrics.** Empirical coverage (marginal and conditional) is the main quantity of interest for prediction sets. We evaluate worst-slab conditional coverage (Cauchois et al., 2020; Romano et al., 2020b) in our experiments, see details in Appendix B.2. We also measure the total size of the predicted sets, scaled by the standard deviation of the response $Y$.

**Experimental setup.** Our experimental setup largely follows the approach outlined in (Wang et al., 2023). Specifically, we split each dataset into training, calibration, and testing sets. A Mixture Density Network (MDN) with 10 components is then trained to approximate the conditional distribution $P_{Y|X}$. For each calibration and test point, we first compute the Gaussian Mixture parameters, forming $\Pi_{Y|X}$, and subsequently draw $M = 5, 20, 50$ samples from these distributions, which yield $\mathcal{R}_z(x, t)$. This process is repeated across 50 different random splits of each dataset.

**Results** of experiments for $M = 50$ samples are presented in Figures 3 and 4, additional results are available in Appendix B. All methods achieve the target $1 - \alpha$ marginal coverage, except for $\Pi_{Y|X}$.

Standard conformal prediction fails to maintain the conditional coverage as expected. We can also observe that $\texttt{PCP}$ consistently struggles with conditional coverage. On all the datasets $\texttt{CP}^2\texttt{-PCP}$ provides valid conditional coverage, while $\texttt{CQR}$ fails on $\texttt{blog}$ and $\texttt{temp}$. $\texttt{CHR}$ method shows unstable performance not achieving conditional coverage more often than other methods but sometimes providing narrower predictions sets. Additionally, $\texttt{CP}^2\texttt{-PCP}$ significantly outperforms quantile regression-based methods in terms of size of the prediction sets on $\texttt{bike}$, $\texttt{bio}$ and $\texttt{temp}$ datasets. $\texttt{CPCG}$ is a strong baseline, but our method shows better performance still. For example, on $\texttt{blog}$, $\texttt{fb1}$ and $\texttt{f2}$ datasets conditional coverage of $\texttt{CP}^2\texttt{-PCP}$ is close to nominal, while prediction set size

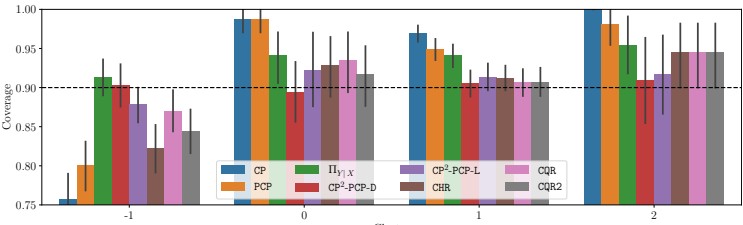

Figure 5: Conditional coverage for different clusters, `fb1` dataset. We have used HDBSCAN algorithm with minimum cluster size of 100, `min_samples` hyper-parameter of 20 and $l_2$ metric. Cluster label $-1$ corresponds to the outliers. Sample size for sampling-based methods was set to 50. Nominal coverage equals $(1 - \alpha) = 0.9$ and is shown by a dashed black line.

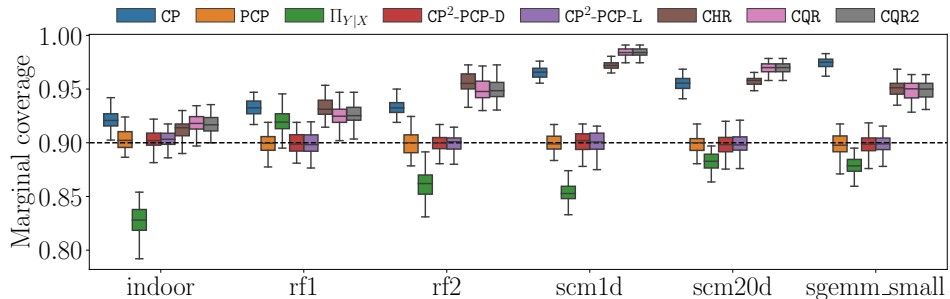

Figure 6: Marginal coverage for multi-target datasets, 50 replications. Sample size was set to 1000. Nominal coverage equals $(1 - \alpha) = 0.9$ and is shown by a dashed black line.

are roughly the same between two methods. On `meps` datasets, conditional coverage is close, while the $CP^2$-PCP sets are smaller. Computational complexity of the prediction is the highest for CPCG, about 10 times that of our approach. LCP shows significantly lower conditional coverage than most methods on larger datasets. CDSplit$^+$ shows good conditional coverage but also large variance between runs. We also noticed that it consistently overcovers marginally, again with large variance. Set sizes for this method are usually small but often comparable to $CP^2$. Overall we consider this method highly unstable on real datasets, requiring additional investigation or hyperparameter tuning.

Finally, we assess conditional coverage with the help of clustering. We apply HDBSCAN (Campello et al., 2013; McInnes & Healy, 2017) method to cluster the test set and then compute coverage within clusters. Results for `fb1` dataset are presented in Figure 5. We again observe that CP and PCP do not achieve conditional coverage and CHR and CQR performance is unstable. $CP^2$–PCP on the other hand maintains valid conditional coverage on all clusters and even on outliers (cluster label $-1$). Note that these are all outliers combined and they may not lie in the same region of the input space.

### 4.3 Real-world Regression Data with Multi-dimensional Targets

We also study `CP2` family of algorithms on the multi-target regression problems. Since selecting the threshold $\tau$ for our methods is not dependent on the number of dimensions in $Y$ their application is straightforward. On the other hand, most other methods are inherently one-dimensional thus require the use of the Bonferroni correction (Dunn, 1961). Each coordinate is treated independently with miscoverage level adjusted to $\alpha/d$, where $d$ is the number of targets. As a result, for quantile regression-based methods prediction sets are formed as a product of the corresponding intervals.

**Datasets.** We consider open-source multidimensional regression datasets: river flow data `rf1` and `rf2` (Xioufis et al., 2012), supply chain management `scm1d` and `scm20d` (Xioufis et al., 2012), indoor localization `indoor` (Torres-Sospedra et al., 2014), GPU computation time `sgemm_small`[2] (Ballester-Ripoll et al., 2019).

---

[2]The full dataset contains 241600 examples. Due to computational constraints we randomly subsample 10000 examples for each replication of our experiment.

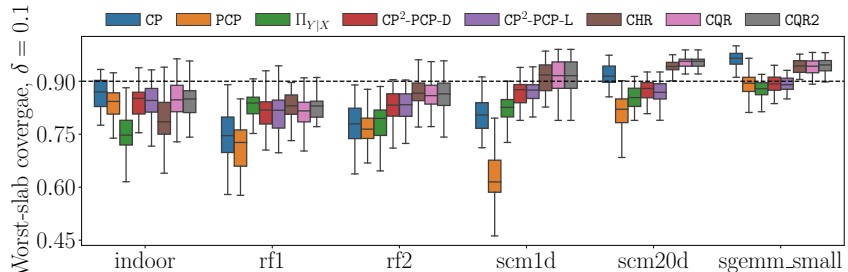

Figure 7: Conditional coverage for multi-target datasets targets, 50 replications. Sample size was set to 1000. Nominal coverage equals $(1 - \alpha) = 0.9$ and is shown in dashed black. Worst-slab coverage parameter $(1 - \delta) = 0.1$.

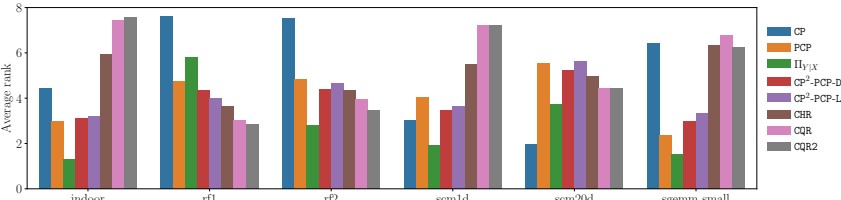

Figure 8: Average rank of the projected set size. For each pair of targets area of the corresponding 2D projection of the prediction set is calculated. For each test point and each pair of targets methods are ranked. Lower rank is smaller area. This graph shows averaged results of 10 replications.

We use the same **metrics** as before: marginal coverage and worst-slab coverage. Evaluating the difference in prediction set size is more complex in case of multiple dimensions. Due to computational constraints we perform pairwise comparisons between our methods and selected baselines, measuring approximate areas of 2D projections of the prediction sets (Wang et al., 2023). These results can be found on Figure 8. We approximate areas using a grid and fewer samples.

Since our methods naturally extend beyond one dimension, the **experimental setup** is almost identical. We use the same underlying model for $P_{Y|X}$, the prediction set is now a union of $d$-dimensional balls of the same radius around the sampled centers. The number of samples is increased to 1000.

**Results.** In Figure 6 we show marginal coverage attained by different algorithms. As expected, naive application of 1D techniques CQR, CQR2 and CHR to multiple outputs produces significant overcover. PCP and $CP^2$ methods naturally extend to multidimensional targets and provide correct marginal coverage. In Figure 7 we present the conditional coverage estimates for multi-target datasets. PCP significantly undercovers on `rf1`, `rf2` and `scm1d` datasets, while $CP^2$ comes very close to the nominal coverage of 0.9. In case of CQR, CQR2 and CHR, they still overcover (`scm20d`, `sgemm`) or perform comparably to our approach. Figure 8 shows the aggregated results of the set size comparisons in multidimensional target setting. For each test point and each pair of axes we rank the methods by the area of the projection of the corresponding prediction set. The plot shows average rank for each method, aggregated across all axes pairs and replications. Lower rank corresponds to smaller area, which is our goal. For datasets `indoor`, `scm1d` and `sgemm_small` our approach performs better, while also providing sharper conditional coverage. On the remaining datasets $CP^2$ performs similarly to the competitors.

## 5   CONCLUSION

We address the challenge of conditional coverage in CP, and overcome previous negative results by assuming the knowledge of a good estimator of $P_{Y|X}$. Our proposed mechanism conformalizes the conditional distribution estimator $\Pi_{Y|X}$ to ensure marginal validity while maintaining approximate conditional coverage guarantees. Specifically, if experts can provide an accurate conditional estimator, our algorithm $CP^2$ generates nearly conditionally valid multidimensional prediction sets. This approach offers a practical solution for tackling heteroscedasticity in machine learning applications.

ACKNOWLEDGEMENTS

This research was partially supported by RSF grant 20-71-10135. V.P. has been supported by a grant of the Lagrange Mathematics and Computing Research Center. E.M. is Funded by the European Union (ERC, Ocean, 101071601). Views and opinions expressed are however those of the author(s) only and do not necessarily reflect those of the European Union or the European Research Council Executive Agency. Neither the European Union nor the granting authority can be held responsible for them.

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

# A  ADDITIONAL RESULTS AND CALCULATIONS

In this section, we analyze the theoretical results of Section 3. First, let's recall the definition of the quantile function for any distribution $\mu_n$ living in $\mathbb{R}$. For any $\alpha \in (0, 1)$, the quantile $Q_{1-\alpha}(\mu_n)$ is defined by

$$Q_{1-\alpha}(\mu_n) = \inf \{t \in \mathbb{R} \colon \mu_n((-\infty, t]) \geq 1 - \alpha\} .$$

Given a measure $\Pi_{Y|X=x}$ defined on $\sigma(\mathcal{Y})$, we consider for all $x \in \mathbb{R}^d$, $z \in \mathcal{Z}$, the parameters $\tau_{x,z}$ and $V_z(x, y)$ given by

$$
\begin{aligned}
\tau_{x,z} &= \inf \left\{\tau \in \mathsf{T} \colon \Pi_{Y|X=x}(\mathcal{R}_z(x; f_\tau(\varphi))) \geq 1 - \alpha\right\} , \\
V_z(x, y) &= \inf \{t \in \mathsf{T} \colon y \in \mathcal{R}_z(x; t)\} ,
\end{aligned}
\tag{8}
$$

where $\varphi$ is chosen as in **H2**, and by convention we set $\inf \emptyset = \infty$. We denote by $\delta_v$ the Dirac measure at $v \in \mathbb{R}$, and write $\tau_k = \tau_{X_k, Z_k}$ and $V_k = V_{Z_k}(X_k, Y_k)$. In this Appendix, we study the coverage of the prediction set given $\forall (x, z) \in \mathbb{R} \times \mathcal{Z}$ by

$$\mathcal{C}_\alpha(x) = \mathcal{R}_z\left(x; f_{\tau_{x,z}}\left(Q_{1-\alpha}(\mu_n)\right)\right) ,$$

where the distribution $\mu_n$ is defined as

$$\mu_n = \frac{1}{n+1} \sum_{k=1}^n \delta_{f_{\tau_k}^{-1}(V_k)} + \frac{1}{n+1} \delta_\infty .$$

The key idea behind the choice of $\tau_k$ is to ensure that the conditional coverage of the prediction set $\mathcal{C}_\alpha(X_k)$ is approximately $1 - \alpha$ when the empirical distribution $\Pi_{Y|X=X_k}$ is close to $P_{Y|X=X_k}$. In other words, $\tau_k$ is chosen such that the probability of the observed value $Y_k$ given $X_k$ falling inside the prediction set $\mathcal{C}_\alpha(X_k)$ is close to $1 - \alpha$. On the other hand, the parameter $V_k$ is used to ensure that the prediction set $\mathcal{R}_{Z_k}(X_k; V_k)$ contains the observed value $Y_k$. Moreover, note that $\tau_k$ only depends on the input data $(X_k, Z_k)$, while $V_k$ depends on $(X_k, Y_k, Z_k)$. Thus, the i.i.d. property of $\{(X_k, Y_k, Z_k) \colon k \in [n+1]\}$ ensures that the $\{(\tau_k, V_k)\}_{k=1}^{n+1}$ are also i.i.d.

## A.1  PROOF OF THEOREMS 3.1 AND 3.2

**Lemma A.1.** *Assume **H**1 hold. For any $(x, y, z) \in \mathbb{R}^d \times \mathcal{Y} \times \mathcal{Z}$, $V_z(x, y)$ exists in $\mathsf{T}$, and we have $y \in \mathcal{R}_z(x; V_z(x, y))$.*

*Proof.* Let $(x, y, z) \in \mathbb{R}^d \times \mathcal{Y} \times \mathcal{Z}$ be fixed. Since $\cap_{t \in \mathsf{T}} \mathcal{R}_z(x; t) = \emptyset$ and $\cup_{t \in \mathsf{T}} \mathcal{R}_z(x; t) = \mathcal{Y}$, we deduce the existence of $t_0$ and $t_1$ such that $y \notin \mathcal{R}_z(x; t_0)$ and $y \in \mathcal{R}_z(x; t_1)$. Therefore, $\{t \in \mathsf{T} \colon y \in \mathcal{R}_z(x; t)\}$ is non-empty and lower-bounded by $t_0$. Thus, the infimum $V_z(x, y)$ exists. Now, let's prove that $y \in \mathcal{R}_z(x; V_z(x, y))$. Since $V_z(x, y) = \inf\{t \in \mathsf{T} \colon y \in \mathcal{R}_z(x; t)\}$, we deduce the existence of a decreasing sequence $\{\lambda_n\}_{n \in \mathbb{N}}$ such that $y \in \mathcal{R}_z(x; \lambda_n)$ and $\lim_{n \to \infty} \lambda_n = V_z(x, y)$. By definition of $\{\lambda_n\}_{n \in \mathbb{N}}$, we have $y \in \cap_{n \in \mathbb{N}} \mathcal{R}_z(x; \lambda_n)$. However, using **H**1, remark that

$$
\begin{aligned}
\cap_{n \in \mathbb{N}} \mathcal{R}_z(x; \lambda_n) &= \cap_{n \in \mathbb{N}} \cap_{t > \lambda_n} \mathcal{R}_z(x; t) \\
&= \cap_{t > \lim_{n \to \infty} \lambda_n} \mathcal{R}_z(x; t) \\
&= \cap_{t > V_z(x,y)} \mathcal{R}_z(x; t) = \mathcal{R}_z(x; V_z(x, y)).
\end{aligned}
$$

Since $y \in \cap_{n \in \mathbb{N}} \mathcal{R}_z(x; \lambda_n)$, it implies that $y \in \mathcal{R}_z(x; V_z(x, y))$.  $\square$

We will now present the proof for Theorem 3.1, which establishes the marginal validity of our proposed method.

**Theorem A.2.** *Assume **H**1-**H**2 hold, if $\{f_{\tau_k}^{-1}(V_k)\}_{k=1}^{n+1}$ are almost surely distinct, then it follows*

$$1 - \alpha \leq \mathbb{P}^{\mathcal{T}}\left(Y_{n+1} \in \mathcal{C}_\alpha(X_{n+1})\right) < 1 - \alpha + \frac{1}{n+1} .
\tag{9}$$

*Proof.* Using Lemma A.1, we have

$$
\begin{aligned}
\mathbb{P}^{\mathcal{T}}\left(Y_{n+1} \in \mathcal{C}_\alpha(X_{n+1})\right) &= \mathbb{P}^{\mathcal{T}}\left(Y_{n+1} \in \mathcal{R}_{Z_{n+1}}\left(X_{n+1}, f_{\tau_{n+1}}(Q_{1-\alpha}(\mu_n))\right)\right) \\
&= \mathbb{P}^{\mathcal{T}}\left(V_{n+1} \leq f_{\tau_{n+1}}(Q_{1-\alpha}(\mu_n))\right) .
\end{aligned}
$$

Since $v \mapsto f_{\tau_{n+1}}(v)$ is increasing by **H2**, we deduce that

$$\mathbb{P}^{\mathcal{T}} \left( V_{n+1} \leq f_{\tau_{n+1}}(Q_{1-\alpha}(\mu_n)) \right) = \mathbb{P}^{\mathcal{T}} \left( f_{\tau_{n+1}}^{-1}(V_{n+1}) \leq Q_{1-\alpha}(\mu_n) \right).$$

Denote by $V_k = f_{\tau_k}^{-1}(V_k)$, the exchangeability of the data $\{(X_k, Y_k, Z_k) \colon k \in [n+1]\}$ implies that

$$\mathbb{P}^{\mathcal{T}} \left( V_{n+1} \leq Q_{1-\alpha} \left( \sum_{k=1}^{n} \frac{\delta_{V_k}}{n+1} + \frac{\delta_\infty}{n+1} \right) \right) = \mathbb{P}^{\mathcal{T}} \left( V_{n+1} \leq Q_{1-\alpha} \left( \sum_{k=1}^{n+1} \frac{\delta_{V_k}}{n+1} \right) \right)$$

$$= \frac{1}{n+1} \sum_{k=1}^{n+1} \mathbb{E}^{\mathcal{T}} \left[ \mathbb{1}_{V_k} \leq Q_{1-\alpha} \left( \frac{1}{n+1} \sum_{k=1}^{n+1} \delta_{V_k} \right) \right]$$

$$= \mathbb{E}^{\mathcal{T}} \left[ \mathbb{E}^{\mathcal{T}} \left[ \mathbb{1}_{V_I} \leq Q_{1-\alpha} \left( \frac{1}{n+1} \sum_{k=1}^{n+1} \delta_{V_k} \right) \, \Big| \, V_1, \dots, V_{n+1} \right] \right],$$

where $I \sim \mathcal{U}nif(1, \dots, n+1)$. Therefore, the definition of the quantile function implies the lower bound in (9). Moreover, if there are no ties between the $\{V_k\}_{k=1}^{n+1}$, then

$$\mathbb{P}^{\mathcal{T}} \left( f_{\tau_{n+1}}^{-1}(V_{n+1}) \leq Q_{1-\alpha}(\mu_n) \right) < 1 - \alpha + \frac{1}{n+1}.$$

$\square$

The following lemma provides conditions under which $\Pi_{Y|X=x}(\mathcal{R}_z(x; f_{\tau_{x,z}}(\varphi))) \geq 1 - \alpha$.

**Lemma A.3.** *Assume **H1**-**H2** hold, and let $\alpha \in (0, 1)$, $x \in \mathbb{R}^d$, $z \in \mathcal{Z}$. If $\Pi_{Y|X=x}$ is a probability measure, then $\tau_{x,z}$ is defined in $\mathsf{T}$ and $\Pi_{Y|X=x}(\mathcal{R}_z(x; f_{\tau_{x,z}}(\varphi))) \geq 1 - \alpha$.*

*Proof.* Let $x \in \mathbb{R}^d$ be such that $\Pi_{Y|X=x}$ is a probability measure, and fix $z \in \mathcal{Z}$. Since $\tau \mapsto f_\tau(\varphi)$ is increasing and bijective by **H2**, we have

$$\sup_{\tau \in \mathsf{T}} \Pi_{Y|X=x}(\mathcal{R}_z(x; f_\tau(\varphi))) = \Pi_{Y|X=x} \left( \cup_{\tau \in \mathsf{T}} \mathcal{R}_z(x; f_\tau(\varphi)) \right)$$

$$= \Pi_{Y|X=x} \left( \cup_{t \in \mathsf{T}} \mathcal{R}_z(x; t) \right) = 1.$$

The previous equality shows the existence of $\tau \in \mathsf{T}$ such that $\Pi_{Y|X=x}(\mathcal{R}_z(x; f_\tau(\varphi))) \geq 1 - \alpha$. Therefore $\{\tau \in \mathsf{T} \colon \Pi_{Y|X=x}(\mathcal{R}_z(x; f_\tau(\varphi))) \geq 1 - \alpha\}$ is non-empty. This proves the existence of $\tau_{x,z} = \inf\{\tau \in \mathsf{T} \colon \Pi_{Y|X=x}(\mathcal{R}_z(x; f_\tau(\varphi))) \geq 1 - \alpha\}$ in $\mathsf{T} \cup \{-\infty\}$. Moreover, $\tau_{x,z} > -\infty$, otherwise we would have

$$1 - \alpha \leq \inf_{\tau \in \mathsf{T}} \Pi_{Y|X=x}(\mathcal{R}_z(x; f_\tau(\varphi))) = \Pi_{Y|X=x} \left( \cap_{t \in \mathsf{T}} \mathcal{R}_z(x; t) \right) = 0.$$

Therefore, we deduce that $\tau_{x,z} \in \mathsf{T}$. Lastly, remark that

$$\Pi_{Y|X=x}(\mathcal{R}_z(x; f_{\tau_{x,z}}(\varphi))) = \Pi_{Y|X=x}(\cap_{\tau > \tau_{x,z}} \mathcal{R}_z(x; f_\tau(\varphi)))$$

$$= \inf_{\tau > \tau_{x,z}} \Pi_{Y|X=x}(\mathcal{R}_z(x; f_\tau(\varphi))) \geq 1 - \alpha.$$

$\square$

Now, we prove Theorem 3.2. This result guarantees that the conditional confidence intervals constructed by our method approximately satisfy the desired coverage of $1-\alpha$. Given $(x, y) \in \mathbb{R}^d \times \mathcal{Z}$, let's introduce

$$p_{n+1}(x, z) = \mathbb{P}^{\mathcal{T}} \left( Q_{1-\alpha}(\mu_n) < f_{\tau_{x,z}}^{-1}(V_z(x, Y_{n+1})) \leq \varphi \, | \, X_{n+1} = x, Z_{n+1} = z \right),$$

$$q_{n+1}(x, z) = \mathbb{P}^{\mathcal{T}} \left( \varphi < f_{\tau_{x,z}}^{-1}(V_z(x, Y_{n+1})) \leq Q_{1-\alpha}(\mu_n) \, | \, X_{n+1} = x, Z_{n+1} = z \right).$$

**Theorem A.4.** *Assume **H1-H2** hold, let $x \in \mathbb{R}^d$ be such that $\Pi_{Y|X=x}$ is a probability measure. For any $z \in \mathcal{Z}$, it follows that*

$$1-\alpha-\mathrm{d_{TV}}(\mathrm{P}_{Y|X=x};\Pi_{Y|X=x})-p_{n+1}(x,z) \leq \mathbb{P}^{\mathcal{T}}\left(Y_{n+1} \in \mathcal{C}_\alpha(X_{n+1}) \mid X_{n+1} = x, Z_{n+1} = z\right)$$
$$\leq \Pi_{Y|X=x}(\mathcal{R}_z(x; f_{\tau_{x,z}}(\varphi))) + \mathrm{d_{TV}}(\mathrm{P}_{Y|X=x};\Pi_{Y|X=x}) + q_{n+1}(x,z).$$

*Proof.* First, recall that $\mathcal{C}_\alpha(x)$ is given in (6), and $V_z(x, Y_{n+1})$ is defined in (8). Applying Lemma A.1, we know that $V_z(x, Y_{n+1})$ is defined in $\mathsf{T}$, and also that $Y_{n+1} \in \mathcal{R}_z(x; V_z(x, Y_{n+1}))$. Hence, it holds

$$\mathbb{P}^{\mathcal{T}}\left(Y_{n+1} \in \mathcal{C}_\alpha(X_{n+1}) \mid X_{n+1} = x, Z_{n+1} = z\right)$$
$$= \mathbb{P}^{\mathcal{T}}\left(Y_{n+1} \in \mathcal{R}_z\left(x; f_{\tau_{x,z}}(Q_{1-\alpha}(\mu_n))\right) \mid X_{n+1} = x, Z_{n+1} = z\right)$$
$$= \mathbb{P}^{\mathcal{T}}\left(V_z(x, Y_{n+1}) \leq f_{\tau_{x,z}}\left(Q_{1-\alpha}(\mu_n)\right) \mid X_{n+1} = x, Z_{n+1} = z\right). \quad (10)$$

Let's introduce the term $\mathbb{P}^{\mathcal{T}}(V_z(x, Y_{n+1}) \leq f_{\tau_{x,z}}(\varphi) \mid X_{n+1} = x, Z_{n+1} = z)$ as follows:

$$\mathbb{P}^{\mathcal{T}}\left(V_z(x, Y_{n+1}) \leq f_{\tau_{x,z}}\left(Q_{1-\alpha}(\mu_n)\right) \mid X_{n+1} = x, Z_{n+1} = z\right)$$
$$= \mathbb{P}^{\mathcal{T}}\left(V_z(x, Y_{n+1}) \leq f_{\tau_{x,z}}\left(Q_{1-\alpha}(\mu_n)\right) \mid X_{n+1} = x, Z_{n+1} = z\right)$$
$$\pm \mathbb{P}^{\mathcal{T}}\left(V_z(x, Y_{n+1}) \leq f_{\tau_{x,z}}(\varphi) \mid X_{n+1} = x, Z_{n+1} = z\right). \quad (11)$$

Now, we will control the difference between the two terms of the previous equation. Let $A$ and $B$ be defined as

$$A = \mathbb{P}^{\mathcal{T}}\left(f_{\tau_{x,z}}^{-1}(V_z(x, Y_{n+1})) \leq Q_{1-\alpha}(\mu_n) < \varphi \mid X_{n+1} = x, Z_{n+1} = z\right),$$
$$B = \mathbb{P}^{\mathcal{T}}\left(f_{\tau_{x,z}}^{-1}(V_z(x, Y_{n+1})) \leq \varphi \leq Q_{1-\alpha}(\mu_n) \mid X_{n+1} = x, Z_{n+1} = z\right).$$

We have

$$\mathbb{P}^{\mathcal{T}}\left(V_z(x, Y_{n+1}) \leq f_{\tau_{x,z}}\left(Q_{1-\alpha}(\mu_n)\right) \mid X_{n+1} = x, Z_{n+1} = z\right)$$
$$= A + B + \mathbb{P}^{\mathcal{T}}\left(\varphi < f_{\tau_{x,z}}^{-1}(V_z(x, Y_{n+1})) \leq Q_{1-\alpha}(\mu_n) \mid X_{n+1} = x, Z_{n+1} = z\right),$$

and also

$$\mathbb{P}^{\mathcal{T}}\left(V_z(x, Y_{n+1}) \leq f_{\tau_{x,z}}(\varphi) \mid X_{n+1} = x, Z_{n+1} = z\right)$$
$$= A + B + \mathbb{P}^{\mathcal{T}}\left(Q_{1-\alpha}(\mu_n) < f_{\tau_{x,z}}^{-1}(V_z(x, Y_{n+1})) \leq \varphi \mid X_{n+1} = x, Z_{n+1} = z\right).$$

Therefore, the difference between the terms introduced in (11) can be rewritten as

$$\mathbb{P}^{\mathcal{T}}\left(V_z(x, Y_{n+1}) \leq f_{\tau_{x,z}}\left(Q_{1-\alpha}(\mu_n)\right) \mid X_{n+1} = x, Z_{n+1} = z\right)$$
$$- \mathbb{P}^{\mathcal{T}}\left(V_z(x, Y_{n+1}) \leq f_{\tau_{x,z}}(\varphi) \mid X_{n+1} = x, Z_{n+1} = z\right)$$
$$= \mathbb{P}^{\mathcal{T}}\left(\varphi < f_{\tau_{x,z}}^{-1}(V_z(x, Y_{n+1})) \leq Q_{1-\alpha}(\mu_n) \mid X_{n+1} = x, Z_{n+1} = z\right)$$
$$- \mathbb{P}^{\mathcal{T}}\left(Q_{1-\alpha}(\mu_n) < f_{\tau_{x,z}}^{-1}(V_z(x, Y_{n+1})) \leq \varphi \mid X_{n+1} = x, Z_{n+1} = z\right). \quad (12)$$

1. By definition of the total variation distance, we have

$$\mathbb{P}^{\mathcal{T}}\left(V_z(x, Y_{n+1}) \leq f_{\tau_{x,z}}(\varphi) \mid X_{n+1} = x, Z_{n+1} = z\right)$$
$$\geq \mathbb{P}^{\mathcal{T}}\left(V_z(x, \hat{Y}_{n+1}) \leq f_{\tau_{x,z}}(\varphi) \mid X_{n+1} = x, Z_{n+1} = z\right) - \mathrm{d_{TV}}(\mathrm{P}_{Y|X=x};\Pi_{Y|X=x}).$$

Moreover, Lemma A.3 implies that

$$\mathbb{P}^{\mathcal{T}}\left(V_z(x, \hat{Y}_{n+1}) \leq f_{\tau_{x,z}}(\varphi) \mid X_{n+1} = x, Z_{n+1} = z\right)$$
$$= \mathbb{P}^{\mathcal{T}}\left(\hat{Y}_{n+1} \in \{y \in \mathcal{Y} \colon V_z(x,y) \leq f_{\tau_{x,z}}(\varphi)\} \mid X_{n+1} = x, Z_{n+1} = z\right)$$
$$= \mathbb{P}^{\mathcal{T}}\left(\hat{Y}_{n+1} \in \mathcal{R}_z(x; f_{\tau_{x,z}}(\varphi)) \mid X_{n+1} = x, Z_{n+1} = z\right)$$
$$= \Pi_{Y|X=x}(\mathcal{R}_z(x; f_{\tau_{x,z}}(\varphi))) \geq 1 - \alpha.$$

Therefore, we deduce that

$$\mathbb{P}^{\mathcal{T}}\left(V_z(x, Y_{n+1}) \leq f_{\tau_{x,z}}(\varphi) \,|\, X_{n+1} = x, Z_{n+1} = z\right) \geq 1 - \alpha - \mathrm{d}_{\mathrm{TV}}(\mathrm{P}_{Y|X=x}; \Pi_{Y|X=x}).$$

Combining the previous result with (11) and (12) shows that

$$\mathbb{P}^{\mathcal{T}}\left(V_z(x, Y_{n+1}) \leq f_{\tau_{x,z}}\left(Q_{1-\alpha}(\mu_n)\right) \,|\, X_{n+1} = x, Z_{n+1} = z\right) \geq 1 - \alpha - \mathrm{d}_{\mathrm{TV}}(\mathrm{P}_{Y|X=x}; \Pi_{Y|X=x})$$
$$- \mathbb{P}^{\mathcal{T}}\left(Q_{1-\alpha}(\mu_n) < f_{\tau_{x,z}}^{-1}\left(V_z(x, Y_{n+1})\right) \leq \varphi \,|\, X_{n+1} = x, Z_{n+1} = z\right).$$

Finally, using (10) gives a lower bound on $\mathbb{P}^{\mathcal{T}}\left(Y_{n+1} \in \mathcal{C}_\alpha(X_{n+1}) \,|\, X_{n+1} = x, Z_{n+1} = z\right)$.

2. By definition of the total variation distance, we have

$$\mathbb{P}^{\mathcal{T}}\left(V_z(x, Y_{n+1}) \leq f_{\tau_{x,z}}(\varphi) \,|\, X_{n+1} = x, Z_{n+1} = z\right)$$
$$\leq \mathbb{P}^{\mathcal{T}}\left(V_z(x, \hat{Y}_{n+1}) \leq f_{\tau_{x,z}}(\varphi) \,|\, X_{n+1} = x, Z_{n+1} = z\right) + \mathrm{d}_{\mathrm{TV}}(\mathrm{P}_{Y|X=x}; \Pi_{Y|X=x}).$$

Moreover, Lemma A.3 implies that

$$\mathbb{P}^{\mathcal{T}}\left(V_z(x, \hat{Y}_{n+1}) \leq f_{\tau_{x,z}}(\varphi) \,|\, X_{n+1} = x, Z_{n+1} = z\right) = \Pi_{Y|X=x}(\mathcal{R}_z(x; f_{\tau_{x,z}}(\varphi))).$$

Therefore, we deduce that

$$\mathbb{P}^{\mathcal{T}}\left(V_z(x, Y_{n+1}) \leq f_{\tau_{x,z}}(\varphi) \,|\, X_{n+1} = x, Z_{n+1} = z\right)$$
$$\leq \Pi_{Y|X=x}(\mathcal{R}_z(x; f_{\tau_{x,z}}(\varphi))) + \mathrm{d}_{\mathrm{TV}}(\mathrm{P}_{Y|X=x}; \Pi_{Y|X=x}).$$

Finally, combining the previous result with (11) and (12) shows that

$$\mathbb{P}^{\mathcal{T}}\left(V_z(x, Y_{n+1}) \leq f_{\tau_{x,z}}\left(Q_{1-\alpha}(\mu_n)\right) \,|\, X_{n+1} = x, Z_{n+1} = z\right)$$
$$\leq \Pi_{Y|X=x}(\mathcal{R}_z(x; f_{\tau_{x,z}}(\varphi))) + \mathrm{d}_{\mathrm{TV}}(\mathrm{P}_{Y|X=x}; \Pi_{Y|X=x}) + q_{n+1}(x, z).$$

$\square$

## A.2 Bound on $p_{n+1}^{(x,z)}$ and $q_{n+1}^{(x,z)}$

The objective of this section is to study the conditional guarantee obtained in Theorem A.4. Under some assumptions, we have demonstrated that the conditional coverage is controlled as follows:

$$1 - \alpha - \mathrm{d}_{\mathrm{TV}}(\mathrm{P}_{Y|X=x}; \Pi_{Y|X=x}) - p_{n+1}(x, z) \leq \mathbb{P}^{\mathcal{T}}\left(Y_{n+1} \in \mathcal{C}_\alpha(X_{n+1}) \,|\, X_{n+1} = x, Z_{n+1} = z\right)$$
$$\leq \Pi_{Y|X=x}(\mathcal{R}_z(x; f_{\tau_{x,z}}(\varphi))) + \mathrm{d}_{\mathrm{TV}}(\mathrm{P}_{Y|X=x}; \Pi_{Y|X=x}) + q_{n+1}(x, z),$$

In the following, we consider the cumulative density functions $F \colon t \mapsto \mathbb{P}^{\mathcal{T}}(f_{\tau_{X,Z}}^{-1}(V_Z(X, Y)) \leq t)$ and $\hat{F} \colon t \mapsto \mathbb{P}^{\mathcal{T}}(f_{\tau_{X,Z}}^{-1}(V_Z(X, \hat{Y})) \leq t)$, where $(X, Y, Z) \sim \mathrm{P}_X \times \mathrm{P}_{Y|X} \times \Pi_{Z|X}$ and $(X, \hat{Y}, Z) \sim \mathrm{P}_X \times \Pi_{Y|X} \times \Pi_{Z|X}$. We denote by $\mu$ and $\hat{\mu}$ the law of the random variables $f_{\tau_{X,Z}}^{-1}(V_Z(X, Y))$ and $f_{\tau_{X,Z}}^{-1}(V_Z(X, \hat{Y}))$. Moreover, recall that $\mu_n = \frac{1}{n+1}\sum_{k=1}^{n} \delta_{f_{\tau_k}^{-1}(V_k)} + \frac{1}{n+1}\delta_\infty$. Note, the quantile $Q_{1-\alpha}(\mu_n)$ is an order statistic with a known distribution that converges to the true quantile $Q_{1-\alpha}(\mu)$. The quantile is defined for any $t \in (0, 1)$ by

$$Q_t(\nu) = \inf\{u \in \mathbb{R} \colon \nu((-\infty, u]) \geq t\}, \qquad \text{where } \nu \in \{\mu, \mu_n, \hat{\mu}\}. \tag{13}$$

**Theorem A.5.** *Assume **H1**-**H2** hold, and let $x \in \mathbb{R}^d$ be such that $\Pi_{Y|X=x}$ is a probability measure. For any $\epsilon \in [0, 1-\alpha)$, if $p_\epsilon = \mathbb{P}^{\mathcal{T}}(f_{\tau_{X,Z}}^{-1}(V_Z(X, Y)) < Q_{1-\alpha-\epsilon}(\mu)) \leq 1 - \alpha$, then it follows that*

$$p_{n+1}(x, z) \leq \mathbb{P}^{\mathcal{T}}\left(Q_{1-\alpha-\epsilon}(\mu) < f_{\tau_{x,y}}^{-1}(V_z(x, Y_{n+1}) \leq Q_{1-\alpha}(\hat{\mu}) \,|\, X_{n+1} = x, Z_{n+1} = z\right)$$
$$+ \exp\left(-np_\epsilon(1-p_\epsilon)h\left(\frac{1-\alpha-p_\epsilon}{p_\epsilon(1-p_\epsilon)}\right)\right),$$

*where $h \colon u \mapsto (1+u)\log(1+u) - u$.*

*Proof.* Let $\epsilon \in [0, 1 - \alpha)$, $x \in \mathbb{R}^d$, and consider

$$A = \{Q_{1-\alpha}(\mu_n) < Q_{1-\alpha-\epsilon}(\mu)\},$$
$$B_{x,z} = \left\{ y \in \mathcal{Y} \colon f_{\tau_{x,z}}(Q_{1-\alpha-\epsilon}(\mu)) < V_z(x,y) \leq f_{\tau_{x,z}}(\varphi) \right\}.$$

We have

$$\mathbb{P}^{\mathcal{T}} \left( f_{\tau_{x,z}}(Q_{1-\alpha}(\mu_n)) < V_z(x, Y_{n+1} \leq f_{\tau_{x,z}}(\varphi) \,|\, X_{n+1} = x, Z_{n+1} = z \right)$$
$$\leq \mathbb{P}^{\mathcal{T}} \left( A \,|\, X_{n+1} = x, Z_{n+1} = z \right) + \mathbb{P}^{\mathcal{T}} \left( Y_{n+1} \in B_{x,z} \,|\, X_{n+1} = x, Z_{n+1} = z \right).$$

Now, let's upper bound the first term of the right-hand side equation. First, remark that

$$\{Q_{1-\alpha}(\mu_n) < Q_{1-\alpha-\epsilon}(\mu)\} \Leftrightarrow \left\{ \frac{1}{n+1} \sum_{k=1}^{n} \mathbb{1}_{f_{\tau_k}^{-1}(V_k) < Q_{1-\alpha-\epsilon}(\mu)} \geq 1 - \alpha \right\}.$$

Thus, we deduce that

$$\mathbb{P}^{\mathcal{T}} \left( A \,|\, X_{n+1} = x, Z_{n+1} = z \right) \leq \mathbb{P}^{\mathcal{T}} \left( \sum_{k=1}^{n} \mathbb{1}_{f_{\tau_k}^{-1}(V_k) < Q_{1-\alpha-\epsilon}(\mu)} \geq (n+1)(1 - \alpha) \right).$$

Recall that $p_\epsilon = \mathbb{P}^{\mathcal{T}}(f_{\tau_{X,Z}}^{-1}(V_Z(X,Y)) < Q_{1-\alpha-\epsilon}(\mu))$, and also that we assume $p_\epsilon \leq 1 - \alpha$. Therefore, the Bennett's inequality (Boucheron et al., 2003, Theorem 2) implies that

$$\mathbb{P}^{\mathcal{T}} \left( A \,|\, X_{n+1} = x, Z_{n+1} = z \right) \leq \exp\left( -np_\epsilon(1 - p_\epsilon) h\left( \frac{(n+1)(1-\alpha) - np_\epsilon}{np_\epsilon(1 - p_\epsilon)} \right) \right), \quad (14)$$

where $h \colon u \mapsto (1 + u)\log(1 + u) - u$. Moreover, define

$$u_\epsilon = \frac{1 - \alpha - p_\epsilon}{p_\epsilon(1 - p_\epsilon)}, \qquad\qquad \tilde{u}_\epsilon = \frac{(n+1)(1-\alpha) - np_\epsilon}{np_\epsilon(1 - p_\epsilon)}.$$

We have $\tilde{u}_\epsilon \leq u_\epsilon$, from the increasing property of $h$ it follows that

$$\mathbb{P}^{\mathcal{T}} \left( A \,|\, X_{n+1} = x, Z_{n+1} = z \right) \leq \exp\left( -np_\epsilon(1 - p_\epsilon)h(u_\epsilon) \right).$$

Furthermore, the definition of $B_{x,z}$ gives

$$\mathbb{P}^{\mathcal{T}} \left( Y_{n+1} \in B_{x,z} \,|\, X_{n+1} = x, Z_{n+1} = z \right)$$
$$= \mathbb{P}^{\mathcal{T}} \left( f_{\tau_{x,z}}(Q_{1-\alpha-\epsilon}(\mu)) < V_z(x, Y_{n+1} \leq f_{\tau_{x,z}}(\varphi) \,|\, X_{n+1} = x, Z_{n+1} = z \right).$$

Moreover, for any $t \in (-\infty, \varphi)$, we have

$$\hat{F}(t) = \mathbb{P}^{\mathcal{T}} \left( f_{\tau_{X,z}}^{-1}(V_Z(X, \hat{Y})) \leq t \right)$$
$$= \int \mathbb{P}^{\mathcal{T}} \left( f_{\tau_{X,z}}^{-1}(V_Z(X, \hat{Y})) \leq t \,\middle|\, X = x, Z = z \right) \bar{\Pi}_{Z|X=x}(\mathrm{d}z) \, \mathrm{P}_X(\mathrm{d}x)$$
$$= \int \mathbb{P}^{\mathcal{T}} \left( \hat{Y} \in \mathcal{R}\left( x, f_{\tau_{z,z}}(t) \right) \,\middle|\, X = x, Z = z \right) \bar{\Pi}_{Z|X=x}(\mathrm{d}z) \, \mathrm{P}_X(\mathrm{d}x).$$

Using **H2**, the bijective property of $\tau \mapsto f_\tau(\varphi)$ implies the existence of $\nu \in \mathsf{T}$, such that $f_\nu(\varphi) = f_{\tau_{z,z}}(t)$. Note that, $\nu < \tau_{x,z}$ otherwise it would lead to $f_\nu(\varphi) \geq f_{\tau_{x,z}}(\varphi) > f_{\tau_{x,z}}(t)$. The definition of $\tau_{x,z}$ shows that

$$\mathbb{P}^{\mathcal{T}} \left( \hat{Y} \in \mathcal{R}\left( x, f_\nu(\varphi) \right) \,\middle|\, X = x, Z = z \right) < 1 - \alpha.$$

Therefore, we deduce that $Q_{1-\alpha}(\hat{\mu}) \geq \varphi$, and we can conclude that

$$\mathbb{P}^{\mathcal{T}} \left( Y_{n+1} \in B_{x,z} \,|\, X_{n+1} = x, Z_{n+1} = z \right)$$
$$\leq \mathbb{P}^{\mathcal{T}} \left( Q_{1-\alpha-\epsilon}(\mu) < f_{\tau_{x,y}}^{-1}(V_z(x, Y_{n+1}) \leq Q_{1-\alpha}(\hat{\mu}) \,|\, X_{n+1} = x, Z_{n+1} = z \right). \quad (15)$$

Finally, combining (14) and (15) concludes the proof. $\qquad\square$

Given $\alpha \in (0,1)$, define the threshold

$$\epsilon_n = \sqrt{\frac{8\alpha(1-\alpha)\log n}{n}}. \tag{16}$$

**Lemma A.6.** *If the distribution of $f_{\tau_{X,Z}}^{-1}(V_Z(X,Y))$ is continuous, then for all $\epsilon \in [0, 1-\alpha)$, we have $p_\epsilon = \mathbb{P}^{\mathcal{T}}(f_{\tau_{X,Z}}^{-1}(V_Z(X,Y)) < Q_{1-\alpha-\epsilon}(\mu)) = 1 - \alpha - \epsilon$. Moreover, if $\epsilon_n \le \frac{\alpha(1-\alpha)}{8}$, then it follows*

$$\exp\left(-np_{\epsilon_n}(1-p_{\epsilon_n})h\left(\frac{1-\alpha-p_{\epsilon_n}}{p_{\epsilon_n}(1-p_{\epsilon_n})}\right)\right) \le \frac{1}{n},$$

*where $h\colon u \mapsto (1+u)\log(1+u) - u$.*

*Proof.* First, recall that $Q_{1-\alpha-\epsilon}(\mu)$ is defined in (13). If the distribution of $f_{\tau_{X,Z}}^{-1}(V_Z(X,Y))$ is continuous, then we have

$$1 - \alpha - \epsilon \le F(Q_{1-\alpha-\epsilon}(\mu)) = \sup_{\delta>0} F(Q_{1-\alpha-\epsilon}(\mu) - \delta)$$
$$\le \mathbb{P}^{\mathcal{T}}\left(f_{\tau_{X,Z}}^{-1}(V_Z(X,Y)) < Q_{1-\alpha-\epsilon}(\mu)\right) = p_\epsilon \le 1 - \alpha - \epsilon.$$

Therefore, we deduce that $p_\epsilon = 1 - \alpha - \epsilon$. Let's denote

$$\delta_n = (n+1)(1-\alpha) - np_{\epsilon_n}, \qquad u_n = \frac{(n+1)(1-\alpha) - np_{\epsilon_n}}{np_{\epsilon_n}(1-p_{\epsilon_n})}.$$

For any $u \ge 0$, remark that $\log(1+u) \ge u - u^2/2$. Thus, we deduce

$$np_{\epsilon_n}(1-p_{\epsilon_n})\, h\,(u_n) \ge \delta_n \frac{(1+u_n)\log(1+u_n) - u_n}{u_n}$$
$$\ge \delta_n \frac{u_n(1-u_n)}{2}. \tag{17}$$

Now, let's show that $u_n \le 1/4$. We have

$$u_n = \frac{(n+1)(1-\alpha) - np_{\epsilon_n}}{np_{\epsilon_n}(1-p_{\epsilon_n})}$$
$$= \frac{1-\alpha}{np_{\epsilon_n}(1-p_{\epsilon_n})} + \frac{1-\alpha-p_{\epsilon_n}}{p_{\epsilon_n}(1-p_{\epsilon_n})}$$
$$= \frac{1-\alpha}{n(\alpha+\epsilon_n)(1-\alpha-\epsilon_n)} + \frac{\epsilon_n}{(\alpha+\epsilon_n)(1-\alpha-\epsilon_n)}.$$

Therefore, $u_n \le 1/4$ if and only if

$$\frac{1-\alpha}{n} + \epsilon_n \le \frac{(\alpha+\epsilon_n)(1-\alpha-\epsilon_n)}{4}.$$

The function $\epsilon \in [0, 1/2-\alpha] \mapsto (\alpha+\epsilon)(1-\alpha-\epsilon)$ is increasing. Since $\epsilon_n \le \alpha(1-\alpha)/8 \le 1/2 - \alpha$, it is sufficient to prove that

$$\frac{1-\alpha}{n} + \epsilon_n \le \frac{\alpha(1-\alpha)}{4}.$$

Since $\epsilon_n \le \alpha(1-\alpha)/8$, we just need to show that

$$\frac{1-\alpha}{n} \le \frac{\alpha(1-\alpha)}{8}, \qquad \text{i.e.,} \qquad \frac{8\alpha(1-\alpha)}{n} \le \alpha^2(1-\alpha). \tag{18}$$

Again, using the fact that $\epsilon_n \le \alpha(1-\alpha)/8$, we deduce that

$$\frac{8\alpha(1-\alpha)}{n} = \frac{\epsilon_n^2}{\log n} \le \frac{\alpha^2(1-\alpha)^2}{8\log n} = \alpha^2(1-\alpha) \times \frac{(1-\alpha)}{8\log n}.$$

Since $\frac{(1-\alpha)}{8 \log n} \leq 1$, we deduce that (18) holds. This concludes that $u_n \leq 1/4$. Moreover, for any $u \in [0, 0.25]$, we have

$$\delta_n \frac{u(1-u)}{2} \geq \frac{u\delta_n}{4}.$$

Plugging the previous line in (17) implies that

$$\exp\left(-np_{\epsilon_n}(1-p_{\epsilon_n})\,h\,(u_n)\right) \leq \exp\left(-\frac{[(n+1)(1-\alpha) - np_{\epsilon_n}]^2}{4np_{\epsilon_n}(1-p_{\epsilon_n})}\right)$$

$$\leq \exp\left(-\frac{(1-\alpha + n\epsilon_n)^2}{4n(\alpha + \epsilon_n)(1-\alpha - \epsilon_n)}\right)$$

$$\leq \exp\left(-\frac{n\epsilon_n^2}{4(\alpha + \epsilon_n)(1-\alpha - \epsilon_n)}\right). \tag{19}$$

Lastly, since $\epsilon_n \leq \alpha$, it follows that

$$\frac{n\epsilon_n^2}{4(\alpha + \epsilon_n)(1-\alpha - \epsilon_n)} = \frac{2\alpha(1-\alpha)\log n}{(\alpha + \epsilon_n)(1-\alpha - \epsilon_n)} \geq \log n.$$

Combining the previous line with (19) completes the proof. □

For any $\epsilon \in [0, \alpha)$, define

$$q_\epsilon = \mathbb{P}^{\mathcal{T}}\left(f_{\tau_{X,Z}}^{-1}(V_Z(X,Y)) < Q_{1-\alpha+\epsilon}(\mu)\right).$$

**Theorem A.7.** *Assume **H**1-**H**2 hold, and let $x \in \mathbb{R}^d$ be such that $\Pi_{Y|X=x}$ is a probability measure. If the distribution of $f_{\tau_{X,Z}}^{-1}(V_Z(X,Y))$ is continuous and $n^{-1}\log n \leq 8^{-3}\alpha(1-\alpha)$, then, it holds*

$$q_{n+1}(x,z) \leq \frac{1}{n} + \mathbb{P}^{\mathcal{T}}\left(Q_{1-\alpha}(\hat{\mu}) < f_{\tau_{x,y}}^{-1}(V_z(x,Y_{n+1}) \leq Q_{1-\alpha+\epsilon_n}(\mu) \mid X_{n+1} = x, Z_{n+1} = z\right), \tag{20}$$

*where $\epsilon_n$ is defined in (16).*

*Proof.* Let's consider

$$A = \{Q_{1-\alpha+\epsilon_n}(\mu) < Q_{1-\alpha}(\mu_n)\},$$
$$B_{x,z} = \left\{y \in \mathcal{Y} \colon f_{\tau_{x,z}}(Q_{1-\alpha}(\hat{\mu})) < V_z(x,y) \leq f_{\tau_{x,z}}(Q_{1-\alpha+\epsilon_n}(\mu))\right\}.$$

We have

$$\mathbb{P}^{\mathcal{T}}\left(f_{\tau_{x,z}}(Q_{1-\alpha}(\hat{\mu})) < V_z(x,Y_{n+1} \leq f_{\tau_{x,z}}(Q_{1-\alpha}(\mu_n)) \mid X_{n+1} = x, Z_{n+1} = z\right)$$
$$\leq \mathbb{P}^{\mathcal{T}}\left(A \mid X_{n+1} = x, Z_{n+1} = z\right) + \mathbb{P}^{\mathcal{T}}\left(Y_{n+1} \in B_{x,z} \mid X_{n+1} = x, Z_{n+1} = z\right). \tag{21}$$

Now, let's upper bound the first term of the right-hand side equation. First, remark that

$$\{Q_{1-\alpha+\epsilon_n}(\mu) < Q_{1-\alpha}(\mu_n)\} \Leftrightarrow \left\{\frac{1}{n+1}\sum_{k=1}^n \mathbb{1}_{f_{\tau_k}^{-1}(V_k) < Q_{1-\alpha+\epsilon_n}(\mu)} < 1-\alpha\right\}.$$

Thus, we deduce that

$$\mathbb{P}^{\mathcal{T}}\left(A \mid X_{n+1} = x, Z_{n+1} = z\right) \leq \mathbb{P}^{\mathcal{T}}\left(\sum_{k=1}^n \mathbb{1}_{f_{\tau_k}^{-1}(V_k) < Q_{1-\alpha+\epsilon_n}(\mu_n)} < (n+1)(1-\alpha)\right).$$

Recall that $q_{\epsilon_n} = \mathbb{P}^{\mathcal{T}}(f_{\tau_{X,Z}}^{-1}(V_Z(X,Y)) < Q_{1-\alpha+\epsilon_n}(\mu))$, and also that $q_{\epsilon_n} < 1$ since the distribution of $f_{\tau_{X,Z}}^{-1}(V_Z(X,Y))$ is continuous with $1 - \alpha + \epsilon_n < 1$. Therefore, the Bennett's inequality (Boucheron et al., 2003, Theorem 2) implies that

$$\mathbb{P}^{\mathcal{T}}\left(A \mid X_{n+1} = x, Z_{n+1} = z\right) \leq \exp\left(-nq_{\epsilon_n}(1-q_{\epsilon_n})\,h\left(\frac{(n+1)(1-\alpha) - nq_{\epsilon_n}}{nq_{\epsilon_n}(1-q_{\epsilon_n})}\right)\right),$$

where $h : u \mapsto (1+u)\log(1+u) - u$. Moreover, define

$$u_{\epsilon_n} = \frac{1 - \alpha - q_{\epsilon_n}}{q_{\epsilon_n}(1 - q_{\epsilon_n})}, \qquad \tilde{u}_{\epsilon_n} = \frac{(n+1)(1-\alpha) - nq_{\epsilon_n}}{nq_{\epsilon_n}(1 - q_{\epsilon_n})}.$$

We have $\tilde{u}_{\epsilon_n} \leq u_{\epsilon_n}$, from the increasing property of $h$ combined with Lemma A.6, it follows that

$$\mathbb{P}^{\mathcal{T}}\left(A \mid X_{n+1} = x, Z_{n+1} = z\right) \leq \exp\left(-nq_{\epsilon_n}(1 - q_{\epsilon_n})h(u_{\epsilon_n})\right) \leq n^{-1}.$$

The previous inequality combined with (21) concludes the proof. $\qquad\square$

### A.3 Proof of Theorem 3.3

**Oracle asymptotic conditional coverage.** Before proving the result, we start by briefly discussing the asymptotic conditional coverage guarantee. Assuming the availability of an oracle for the predictive distribution, i.e., $P_{Y|X=x} = \Pi_{Y|X=x}$, we get under **H1** and **H2**, that for any $t \in \mathbb{R}$,

$$\mathbb{P}\left(V_Z(X,Y) \leq f_{\tau_{X,z}}(t) \mid X = x, Z = z\right) = \mathbb{P}\left(Y \in \mathcal{R}_z(x; f_{\tau_{x,z}}(t)) \mid X = x, Z = z\right)$$
$$= \Pi_{Y|X=x}\left(\mathcal{R}_z(x; f_{\tau_{x,z}}(t))\right),$$

where $(X, Y, Z)$ follows the same distribution than $(X_k, Y_k, Z_k)$, $k \in \{1, \dots, n\}$. Note that $\Pi_{Y|X=x}(\mathcal{R}_z(x; f_{\tau_{x,z}}(t))) \geq 1 - \alpha$ if and only if $t \geq \varphi$, which implies that

$$\mathbb{P}(f_{\tau_{X,Z}}^{-1}(V_Z(X,Y)) \leq t \mid (X, Z) = (x, z)) \geq 1 - \alpha \quad \text{if and only if} \quad t \geq \varphi. \tag{22}$$

From (22) it is easily seen that the $(1-\alpha)$-quantile of $f_{\tau_{X,Z}}^{-1}(V_Z(X,Y))$ is $\varphi$. The Glivenko–Cantelli Theorem (Van der Vaart, 2000, Theorem 19.1) demonstrates that $\sup_{t \in \mathbb{R}} |\mu_n(-\infty, t] - \mathbb{P}(f_{\tau_{X,Z}}^{-1}(V_Z(X,Y)) \leq t)| \to 0$ almost surely as $n \to \infty$, where $\mu_n$ is defined in (7). Since the convergence of the c.d.f. implies the convergence of the quantile function (Van der Vaart, 2000, Lemma 21.2), we deduce that $Q_{1-\alpha}(\mu_n) \to \varphi$ almost-surely as $n \to \infty$. Under weak additional conditions this implies that $\lim_{n \to \infty} p_{n+1}(x, z) = 0$, $\bar{\Pi}_{Z|X} \times P_X$-almost everywhere, where $\bar{\Pi}_{Z|X}$ is the distribution used to draw the auxiliary variables $z$; see Appendix A.4. In this case, Theorem 3.1 implies the asymptotic validity of $\mathrm{CP}^2$.

Now we can prove a precise result that takes into account that only an estimate $\Pi_{Y|X}$ of the conditional distribution $P_{Y|X=x}$ is available.

**Theorem A.8.** *Assume **H1**-**H2** and suppose the distributions of $f_{\tau_{X,Z}}^{-1}(V_Z(X,Y))$ and $f_{\tau_{X,Z}}^{-1}(V_Z(X,\hat{Y}))$ are continuous. For any $\alpha \in (0,1)$ and $\rho > 0$, it holds*

$$\mathbb{P}^{\mathcal{T}}\left(\left|\mathbb{P}^{\mathcal{T}}\left(Y_{n+1} \in \mathcal{C}_\alpha(X_{n+1}) \mid X_{n+1}, Z_{n+1}\right) - 1 + \alpha\right| > \rho\right)$$
$$\leq \frac{2n^{-1} + \sqrt{128\alpha(1-\alpha)n^{-1}\log n} + 4\mathrm{d}_{\mathrm{TV}}(P_{X,Y}; P_X \times \Pi_{Y|X})}{\rho}.$$

*Proof.* Let $\rho > 0$ be fixed. Applying Theorem 3.2, we obtain that

$$1 - \alpha - \mathrm{d}_{\mathrm{TV}}(P_{Y|X=x}; \Pi_{Y|X=x}) - p_{n+1}(x, z) \leq \mathbb{P}^{\mathcal{T}}\left(Y_{n+1} \in \mathcal{C}_\alpha(X_{n+1}) \mid X_{n+1} = x, Z_{n+1} = z\right)$$
$$\leq \Pi_{Y|X=x}(\mathcal{R}_z(x; f_{\tau_{x,z}}(\varphi))) + \mathrm{d}_{\mathrm{TV}}(P_{Y|X=x}; \Pi_{Y|X=x}) + q_{n+1}(x, z). \tag{23}$$

**Step 1: Lower bound.** Using the Markov's inequality implies that

$$\mathbb{P}^{\mathcal{T}}\left(\mathbb{P}^{\mathcal{T}}\left(Y_{n+1} \in \mathcal{C}_\alpha(X_{n+1}) \mid X_{n+1}, Z_{n+1}\right) < 1 - \alpha - \rho\right) \leq \mathbb{P}^{\mathcal{T}}\left(\mathrm{d}_{\mathrm{TV}}(P_{Y|X}; \Pi_{Y|X}) + p_{n+1}(X, Z) < \rho\right)$$
$$\leq \frac{\mathbb{E}^{\mathcal{T}}\left[\mathrm{d}_{\mathrm{TV}}(P_{Y|X}; \Pi_{Y|X})\right] + \mathbb{E}^{\mathcal{T}}\left[p_{n+1}(X, Z)\right]}{\rho}. \tag{24}$$

Moreover, using Theorem A.5 with $\Phi(\epsilon) = \epsilon[(u_\epsilon^{-1} - 1)\log(1 + u_\epsilon) - 1]$ and $u_\epsilon = \epsilon(\alpha + \epsilon)^{-1}(1 - \alpha - \epsilon)$, it holds

$$\mathbb{E}^{\mathcal{T}}\left[p_{n+1}(X, Z)\right] = \mathbb{P}^{\mathcal{T}}\left(Q_{1-\alpha}(\mu_n) < f_{\tau_{X,Z}}^{-1}(V_Z(X,Y)) \leq \varphi\right)$$
$$\leq \exp\left(-n\Phi(\epsilon)\right) + \mathbb{P}^{\mathcal{T}}\left(Q_{1-\alpha-\epsilon}(\mu) < f_{\tau_{n+1}}^{-1}(V_{n+1}) \leq Q_{1-\alpha}(\hat{\mu}) \mid X_{n+1} = x, Z_{n+1} = z\right).$$

By Lemma A.6, if $n^{-1} \log n \leq 8^{-3} \alpha (1-\alpha)$, then, setting $\epsilon_n = \sqrt{8\alpha(1-\alpha)n^{-1}\log n}$ ensures that $\exp(-n\Phi(\epsilon_n)) \leq n^{-1}$. We assume in the following that $n^{-1}\log n \leq 8^{-3}\alpha(1-\alpha)$, because, if it not the case, the final upper bound obtained at the end of the proof is still valid. Thus, we get

$$\mathbb{E}^{\mathcal{T}}\left[p_{n+1}(X,Z)\right] \leq n^{-1} + \mathbb{P}^{\mathcal{T}}\left(Q_{1-\alpha-\epsilon_n}(\mu) < f_{\tau_{n+1}}^{-1}(V_{n+1}) \leq Q_{1-\alpha}(\hat{\mu})\right). \tag{25}$$

Let's define $\bar{\gamma}$ by

$$\bar{\gamma} = \min(1, 1 - \alpha + d_{\mathrm{TV}}(\mathrm{P}_{X,Y}; \mathrm{P}_X \times \Pi_{Y|X})).$$

We now show that $Q_{1-\alpha}(\hat{\mu}) \leq Q_{\bar{\gamma}}(\mu)$. By continuity of the cumulative density function of $f_{\tau_{X,z}}^{-1}(V_Z(X,\hat{Y}))$, we have

$$1 - \alpha = \mathbb{P}^{\mathcal{T}}\left(f_{\tau_{X,z}}^{-1}(V_Z(X,\hat{Y})) \leq Q_{1-\alpha}(\hat{\mu})\right)$$
$$\geq \mathbb{P}^{\mathcal{T}}\left(f_{\tau_{X,z}}^{-1}(V_Z(X,Y)) \leq Q_{1-\alpha}(\hat{\mu})\right) - d_{\mathrm{TV}}(\mathrm{P}_{X,Y}; \mathrm{P}_X \times \Pi_{Y|X}).$$

Hence, it follows that

$$\mathbb{P}^{\mathcal{T}}\left(f_{\tau_{X,z}}^{-1}(V_Z(X,Y)) \leq Q_{1-\alpha}(\hat{\mu})\right) \leq \bar{\gamma} \leq \mathbb{P}^{\mathcal{T}}\left(f_{\tau_{X,z}}^{-1}(V_Z(X,Y)) \leq Q_{\bar{\gamma}}(\mu)\right).$$

Thus, the previous line implies that $Q_{1-\alpha}(\hat{\mu}) \leq Q_{\bar{\gamma}}(\mu)$. Once again, using the continuity of the distribution of $f_{\tau_{X,z}}^{-1}(V_Z(X,Y))$, we can write

$$\mathbb{P}^{\mathcal{T}}\left(Q_{1-\alpha}(\hat{\mu}) < f_{\tau_{x,y}}^{-1}(V_z(x,Y_{n+1})) \leq Q_{1-\alpha+\epsilon_n}(\mu)\right) \leq \mathbb{P}^{\mathcal{T}}\left(Q_{1-\alpha-\epsilon_n}(\mu) < f_{\tau_{n+1}}^{-1}(V_{n+1}) \leq Q_{\bar{\gamma}}(\mu)\right)$$
$$= F\left(Q_{\bar{\gamma}}(\mu)\right) - F\left(Q_{1-\alpha-\epsilon_n}(\mu)\right) = \bar{\gamma} + \epsilon_n - 1 + \alpha$$
$$= n^{-1} + \sqrt{8\alpha(1-\alpha)n^{-1}\log n} + d_{\mathrm{TV}}(\mathrm{P}_{X,Y}; \mathrm{P}_X \times \Pi_{Y|X}).$$

Plugging the previous inequality inside (25) yields

$$\mathbb{E}^{\mathcal{T}}\left[p_{n+1}(X,Z)\right] \leq n^{-1} + \sqrt{8\alpha(1-\alpha)n^{-1}\log n} + d_{\mathrm{TV}}(\mathrm{P}_{X,Y}; \mathrm{P}_X \times \Pi_{Y|X}).$$

Therefore, (24) implies that

$$\mathbb{P}^{\mathcal{T}}\left(\mathbb{P}^{\mathcal{T}}\left(Y_{n+1} \in \mathcal{C}_\alpha(X_{n+1}) \mid X_{n+1}, Z_{n+1}\right) < 1 - \alpha - \rho\right)$$
$$\leq \frac{n^{-1} + \sqrt{8\alpha(1-\alpha)n^{-1}\log n} + d_{\mathrm{TV}}(\mathrm{P}_{X,Y}; \mathrm{P}_X \times \Pi_{Y|X}) + \mathbb{E}^{\mathcal{T}}\left[d_{\mathrm{TV}}(\mathrm{P}_{Y|X}; \Pi_{Y|X})\right]}{\rho}. \tag{26}$$

**Step 2: Upper bound.** Using (23), we obtain

$$\mathbb{P}^{\mathcal{T}}\left(\mathbb{P}^{\mathcal{T}}\left(Y_{n+1} \in \mathcal{C}_\alpha(X_{n+1}) \mid X_{n+1}, Z_{n+1}\right) > 1 - \alpha + \rho\right)$$
$$\leq \mathbb{P}^{\mathcal{T}}\left(\Pi_{Y|X=X}(\mathcal{R}_Z(X; f_{\tau_{X,z}}(\varphi))) + d_{\mathrm{TV}}(\mathrm{P}_{Y|X=X}; \Pi_{Y|X=X}) + q_{n+1}^{(X,Z)} > 1 - \alpha + \rho\right).$$

The continuity of the distribution of $f_{\tau_{X,z}}^{-1}(V_Z(X,\hat{Y}))$ implies

$$1 - \alpha = \mathbb{P}^{\mathcal{T}}\left(f_{\tau_{X,z}}^{-1}(V_Z(X,\hat{Y})) \leq \varphi\right) = \int \Pi_{Y|X=x}(\mathcal{R}_z(x; f_{\tau_{x,z}}(\varphi))) \bar{\Pi}_{Z|X=x}(\mathrm{d}z) \mathrm{P}_X(\mathrm{d}x).$$

Since $\Pi_{Y|X=x}(\mathcal{R}_z(x; f_{\tau_{x,z}}(\varphi))) \geq 1 - \alpha$, we deduce that $\Pi_{Y|X=x}(\mathcal{R}_z(x; f_{\tau_{x,z}}(\varphi))) = 1 - \alpha$ almost surely. Therefore, using the Markov's inequality gives

$$\mathbb{P}^{\mathcal{T}}\left(\mathbb{P}^{\mathcal{T}}\left(Y_{n+1} \in \mathcal{C}_\alpha(X_{n+1}) \mid X_{n+1}, Z_{n+1}\right) > 1 - \alpha + \rho\right) \leq \frac{\mathbb{E}^{\mathcal{T}}\left[d_{\mathrm{TV}}(\mathrm{P}_{Y|X}; \Pi_{Y|X})\right] + \mathbb{E}^{\mathcal{T}}\left[q_{n+1}^{(X,Z)}\right]}{\rho}. \tag{27}$$

Moreover, applying Theorem A.7 shows that

$$q_{n+1}(x,z) \leq n^{-1} + \mathbb{P}^{\mathcal{T}}\left(Q_{1-\alpha}(\hat{\mu}) < f_{\tau_{x,y}}^{-1}(V_z(x,Y_{n+1})) \leq Q_{1-\alpha+\epsilon_n}(\mu) \mid X_{n+1} = x, Z_{n+1} = z\right). \tag{28}$$

Let's define $\underline{\gamma}$ by

$$\underline{\gamma} = \min(1, 1 - \alpha - \mathrm{d}_{\mathrm{TV}}(\mathrm{P}_{X,Y}; \mathrm{P}_X \times \Pi_{Y|X})).$$

We now show that $Q_{\underline{\gamma}}(\mu) \leq Q_{1-\alpha}(\hat{\mu})$. By continuity of the cumulative density function of $f_{\tau_{X,Z}}^{-1}(V_Z(X, \hat{Y}))$, we have

$$
\begin{aligned}
1 - \alpha &= \mathbb{P}^{\mathcal{T}}\left(f_{\tau_{X,Z}}^{-1}(V_Z(X, \hat{Y})) \leq Q_{1-\alpha}(\hat{\mu})\right) \\
&\leq \mathbb{P}^{\mathcal{T}}\left(f_{\tau_{X,Z}}^{-1}(V_Z(X, Y)) \leq Q_{1-\alpha}(\hat{\mu})\right) + \mathrm{d}_{\mathrm{TV}}(\mathrm{P}_{X,Y}; \mathrm{P}_X \times \Pi_{Y|X}).
\end{aligned}
$$

Hence, it follows that

$$\underline{\gamma} \leq \mathbb{P}^{\mathcal{T}}\left(f_{\tau_{X,Z}}^{-1}(V_Z(X, Y)) \leq Q_{1-\alpha}(\hat{\mu})\right).$$

Thus, we deduce that $Q_{1-\alpha}(\hat{\mu}) \geq Q_{\underline{\gamma}}(\mu)$. Using the continuity of the distribution of $f_{\tau_{X,Z}}^{-1}(V_Z(X, Y))$, we can write

$$
\begin{aligned}
\mathbb{P}^{\mathcal{T}}\left(Q_{1-\alpha}(\hat{\mu}) < f_{\tau_{x,y}}^{-1}(V_z(x, Y_{n+1})) \leq Q_{1-\alpha+\epsilon_n}(\mu)\right) &\leq \mathbb{P}^{\mathcal{T}}\left(Q_{\underline{\gamma}}(\mu) < f_{\tau_{x,y}}^{-1}(V_z(x, Y_{n+1})) \leq Q_{1-\alpha+\epsilon_n}(\mu)\right) \\
= F\left(Q_{1-\alpha+\epsilon_n}(\mu)\right) - F\left(Q_{\underline{\gamma}}(\mu)\right) &= \epsilon_n - 1 + \alpha - \underline{\gamma} \\
&= n^{-1} + \sqrt{8\alpha(1-\alpha)n^{-1}\log n} + \mathrm{d}_{\mathrm{TV}}(\mathrm{P}_{X,Y}; \mathrm{P}_X \times \Pi_{Y|X}).
\end{aligned}
$$

Plugging the previous inequality inside (28) yields

$$\mathbb{E}^{\mathcal{T}}\left[q_{n+1}^{(X,Z)}\right] \leq n^{-1} + \sqrt{8\alpha(1-\alpha)n^{-1}\log n} + \mathrm{d}_{\mathrm{TV}}(\mathrm{P}_{X,Y}; \mathrm{P}_X \times \Pi_{Y|X}).$$

Therefore, (27) implies that

$$
\begin{aligned}
&\mathbb{P}^{\mathcal{T}}\left(\mathbb{P}^{\mathcal{T}}\left(Y_{n+1} \in \mathcal{C}_\alpha(X_{n+1}) \mid X_{n+1}, Z_{n+1}\right) < 1 - \alpha - \rho\right) \\
&\leq \frac{n^{-1} + \sqrt{8\alpha(1-\alpha)n^{-1}\log n} + \mathrm{d}_{\mathrm{TV}}(\mathrm{P}_{X,Y}; \mathrm{P}_X \times \Pi_{Y|X}) + \mathbb{E}^{\mathcal{T}}\left[\mathrm{d}_{\mathrm{TV}}(\mathrm{P}_{Y|X}; \Pi_{Y|X})\right]}{\rho}. \quad (29)
\end{aligned}
$$

**Step 3: Bound on $\mathbb{E}^{\mathcal{T}}\left[\mathrm{d}_{\mathrm{TV}}(\mathrm{P}_{Y|X}; \Pi_{Y|X})\right]$.** Let's denote $\nu_{Y|X=x} = 2^{-1}(\mathrm{P}_{Y|X=x} + \Pi_{Y|X=x})$. Since $\mathrm{P}_{Y|X=x} \ll \nu_{Y|X=x}$ and $\Pi_{Y|X=x} \ll \nu_{Y|X=x}$, there exists two Radon–Nikodym derivatives $g_1(x, \cdot)$ and $g_1(x, \cdot)$ of $\mathrm{P}_{Y|X=x}$ and $\Pi_{Y|X=x}$ with respect to $\nu_{Y|X=x}$. Moreover, $g_1$ and $g_2$ are also the Radon–Nikodym derivatives of $\mathrm{P}_{X,Y}$ and $\mathrm{P}_X \times \Pi_{Y|X}$ with respect to $\mathrm{P}_X \times \nu_{Y|X}$. By definition of the total variation distance, we have

$$
\begin{aligned}
\mathbb{E}^{\mathcal{T}}\left[\mathrm{d}_{\mathrm{TV}}(\mathrm{P}_{Y|X}; \Pi_{Y|X})\right] &= \int \mathrm{d}_{\mathrm{TV}}(\mathrm{P}_{Y|X}; \Pi_{Y|X})\mathrm{P}_X(\mathrm{d}x) \\
&= \frac{1}{2}\int |g_1(x, y) - g_2(x, y)| \,\nu_{Y|X=x}\mathrm{P}_X(\mathrm{d}x) \\
&= \mathrm{d}_{\mathrm{TV}}(\mathrm{P}_{X,Y}; \mathrm{P}_X \times \Pi_{Y|X}). \quad (30)
\end{aligned}
$$

**Step 4: Combination.** Finally, using (26)-(29) and (30), it follows that

$$
\begin{aligned}
&\mathbb{P}^{\mathcal{T}}\left(\left|\mathbb{P}^{\mathcal{T}}\left(Y_{n+1} \in \mathcal{C}_\alpha(X_{n+1}) \mid X_{n+1}, Z_{n+1}\right) - 1 + \alpha\right| > \rho\right) \\
&\leq \frac{2n^{-1} + \sqrt{128\alpha(1-\alpha)n^{-1}\log n} + 4\mathrm{d}_{\mathrm{TV}}(\mathrm{P}_{X,Y}; \mathrm{P}_X \times \Pi_{Y|X})}{\rho}.
\end{aligned}
$$

Note that the proof assumes $n^{-1}\log n \leq 8^{-3}\alpha(1-\alpha)$. To ensure the validity of the previous bound even when this assumption does not hold, we increased the term $\sqrt{32\alpha(1-\alpha)n^{-1}\log n}$ to $\sqrt{128\alpha(1-\alpha)n^{-1}\log n}$. $\qquad \square$

Now we are ready to prove the result of Theorem 3.3.

**Theorem A.9.** *Assume **H1**-**H2**-**H3** hold. If the distributions of $f_{\tau_{X,Z}}^{-1}(V_Z(X,Y))$ and $f_{\tau_{X,Z}}^{-1}(V_Z(X,\hat{Y}))$ are continuous, then, $\forall \epsilon \in (0,1)$ there exists $(\Lambda_n^{(\epsilon)})_{n\in\mathbb{N}}$ such that $\liminf_{n\to\infty}\mathbb{P}((X_{n+1},Z_{n+1})\in\Lambda_n^{(\epsilon)}) \geq 1-\epsilon$ and also*

$$\sup_{(x,z)\in\Lambda_n^{(\epsilon)}} \left|\mathbb{P}^{\mathcal{T}}\left(Y_{n+1}\in\mathcal{C}_\alpha(X_{n+1})\,|\,(X_{n+1},Z_{n+1})=(x,z)\right)-1+\alpha\right| = O_{\mathbb{P}}\left(\sqrt{n^{-1}\log n}+r_n\right).$$

*Proof.* First of all, define the following variables

$$c_{n+1}(x,z) = \left|\mathbb{P}^{\mathcal{T}}\left(Y_{n+1}\in\mathcal{C}_\alpha(X_{n+1})\,|\,(X_{n+1},Z_{n+1})=(x,z)\right)-1+\alpha\right|,$$
$$d_n = \mathrm{d}_{\mathrm{TV}}(\mathrm{P}_{X,Y};\mathrm{P}_X\times\Pi_{Y|X}^{(m_n)}).$$

Applying Theorem A.8, we obtain

$$\mathbb{P}\left(c_{n+1}(X_{n+1},Z_{n+1})>\rho\right) \leq \mathbb{P}\left(d_n>r_n\right) + \mathbb{P}\left(c_{n+1}(X_{n+1},Z_{n+1})>\rho;d_n\leq r_n\right)$$
$$\leq \mathbb{P}\left(d_n>r_n\right) + \mathbb{E}\left[\mathbb{1}_{d_n\leq r_n}\mathbb{P}^{\mathcal{T}}\left(c_{n+1}(X_{n+1},Z_{n+1})>\rho\right)\right]$$
$$\leq \mathbb{P}\left(d_n>r_n\right) + \frac{2n^{-1}+\sqrt{128\alpha(1-\alpha)n^{-1}\log n}+4r_n}{\rho}.$$

Finally, using **H3**, we get $\lim_{n\to\infty}\mathbb{P}(d_n>r_n)=0$. Therefore, for any $\epsilon>0$, there exist $M_\epsilon>0$ and $\tilde{n}_\epsilon\in\mathbb{N}$ such that, $\forall n\geq\tilde{n}_\epsilon$, it holds

$$\mathbb{P}\left(c_{n+1}(X_{n+1},Z_{n+1})>M_\epsilon\times\left(\sqrt{n^{-1}\log n}+r_n\right)\right)\leq\epsilon. \tag{31}$$

Given $\epsilon\in(0,1)$, let's consider the following set

$$\Lambda_n^{(\epsilon)} = \left\{(X_{n+1}(\omega),Z_{n+1}(\omega))\colon \omega\in\Omega, c_{n+1}(X_{n+1},Z_{n+1})(\omega)\leq M_\epsilon\times\left(\sqrt{n^{-1}\log n}+r_n\right)\right\}.$$

Equation (31) implies that

$$\liminf_{n\to\infty}\mathbb{P}\left((X_{n+1},Z_{n+1})\in\Lambda_n^{(\epsilon)}\right)\geq 1-\epsilon,$$

and by definition of $\Lambda_n^{(\epsilon)}$, we also have

$$\sup_{(x,z)\in\Lambda_n^{(\epsilon)}} c_{n+1}(x,z) = O_{\mathbb{P}}\left(\sqrt{n^{-1}\log n}+r_n\right).$$

$\square$

Note that, (31) also shows that

$$\left|\mathbb{P}^{\mathcal{T}}\left(Y_{n+1}\in\mathcal{C}_\alpha(X_{n+1})\,|\,X_{n+1},Z_{n+1}\right)-1+\alpha\right| = O_{\mathbb{P}}\left(n^{-1/2}\sqrt{\log n}+r_n\right).$$

## A.4 Additional results

Let's denote the conditional c.d.f of $f_{\tau_{X,Z}}^{-1}(V_Z(X,Y))$ by

$$F_{x,z}(\cdot) = \int_{\mathbb{R}^d\times\mathcal{Z}} \mathbb{P}\left(f_{\tau_{x,z}}^{-1}(V_z(x,Y))\leq\cdot\,|\,(X,Z)=(x,z)\right)\bar{\Pi}_{Z|X=x}(\mathrm{d}z)\mathrm{P}_X(\mathrm{d}x).$$

**Lemma A.10.** *Assume that $Q_{1-\alpha}(\mu_n)\to\varphi$ almost-surely as $n\to\infty$. If $F_{X,Z}$ is continuous almost-surely, then $\lim_{n\to\infty}p_{n+1}^{(x,z)}=0$, $\bar{\Pi}_{Z|X}\times\mathrm{P}_X$-almost everywhere.*

*Proof.* First, define the following sets:

$$A = \left\{\omega\in\Omega\colon \lim_{n\to\infty}Q_{1-\alpha}(\mu_n(\omega))=\varphi\right\},$$
$$B = \left\{\omega\in\Omega\colon F_{X(\omega),Z(\omega)}\text{ is continuous}\right\}.$$

For all $\omega \in A \cap B$, it holds

$$\lim_{n\to\infty} F_{X(\omega),Z(\omega)}\left(Q_{1-\alpha}\left(\mu_n(\omega)\right) \wedge \varphi\right) = F_{X(\omega),Z(\omega)}\left(\varphi\right).$$

Moreover, note that we can write

$$p_{n+1}^{(x,z)} = F_{x,z}(\varphi) - F_{x,z}(\varphi \wedge Q_{1-\alpha}(\mu_n)).$$

Hence, we deduce that

$$
\begin{aligned}
1 = \mathbb{P}\left(A \cap B\right) &\leq \mathbb{P}\left(\omega \in \Omega: \lim_{n\to\infty} F_{X(\omega),Z(\omega)}\left(Q_{1-\alpha}\left(\mu_n(\omega)\right) \wedge \varphi\right) = F_{X(\omega),Z(\omega)}\left(\varphi\right)\right) \\
&= \mathbb{P}\left(\lim_{n\to\infty} p_{n+1}^{(X,Z)} = 0\right) \\
&= \int_{\mathbb{R}^d \times \mathcal{Z}} \mathbb{P}\left(\lim_{n\to\infty} p_{n+1}^{(x,z)} = 0 \,\middle|\, (X,Z) = (x,z)\right) \mathrm{P}_{Z|X=x}(\mathrm{d}z)\,\mathrm{P}_X(\mathrm{d}x).
\end{aligned}
$$

The last line implies that $p_{n+1}^{(x,z)} \to 0$ almost $\mathrm{P}_{Z|X} \times \mathrm{P}_X$-everywhere. $\qquad\square$

The prediction set, defined in (6), is derived from the $(1-\alpha)$-quantile of the conformity scores $\{f_{\tau_k}^{-1}(V_k)\}_{k=1}^n \cup \{\infty\}$. However, $\{\infty\}$ can be removed from these conformity scores. Inspired by Romano et al. (2019); Sesia & Candès (2020), we prove a corollary of Theorem 3.1. Its result demonstrates the marginal validity of the prediction set defined as

$$\bar{\mathcal{C}}_\alpha(x) = \mathcal{R}_z\left(x; f_{\tau_{x,z}}\left(Q_{(1-\alpha)(1+n^{-1})}\left(\tfrac{1}{n}\sum_{k=1}^n \delta_{f_{\tau_k}^{-1}(V_k)}\right)\right)\right). \tag{32}$$

While the prediction set $\bar{\mathcal{C}}_\alpha(x)$ relies on the quantile of the distribution $\frac{1}{n}\sum_{k=1}^n \delta_{f_{\tau_k}^{-1}(V_k)}$, its proof reveals that this prediction set is equivalent to $\mathcal{C}_\alpha(x)$.

**Corollary A.11.** *Under the same assumptions as in Theorem 3.1, for any $\alpha \in [1/(n+1), 1]$, we have*

$$1 - \alpha \leq \mathbb{P}\left(Y_{n+1} \in \bar{\mathcal{C}}_\alpha(X_{n+1})\right) < 1 - \alpha + \frac{1}{n+1},$$

*where the upper bound only holds if the conformity scores $\{f_{\tau_k}^{-1}(V_k)\}_{k=1}^{n+1}$ are almost surely distinct.*

*Proof.* Let $\alpha \in \mathbb{R}$ such that $(n+1)^{-1} \leq \alpha \leq 1$, and recall that

$$\mu_n = \frac{1}{n+1}\sum_{k=1}^n \delta_{f_{\tau_k}^{-1}(V_k)} + \frac{1}{n+1}\delta_\infty.$$

Since $\alpha \geq (n+1)^{-1}$, the quantile $Q_{1-\alpha}(\mu_n)$ is the $k_\alpha$-th order statistic of $\bar{V}_1, \ldots, \bar{V}_n$, where

$$\bar{V}_k = f_{\tau_k}^{-1}(V_k), \quad \text{and} \quad k_\alpha = \lceil (1-\alpha)(n+1) \rceil.$$

However, $\forall \beta \in \left(\frac{k_\alpha - 1}{n}, \frac{k_\alpha}{n}\right]$, we have

$$Q_\beta\left(\tfrac{1}{n}\sum_{k=1}^n \delta_{\bar{V}_k}\right) = \bar{V}_{(k_\alpha)}.$$

Since $\mathcal{C}_\alpha(X_{n+1}) = \mathcal{R}_{Z_{n+1}}(X_{n+1}; f_{\tau_{n+1}}(\bar{V}_{(k_\alpha)}))$, Theorem 3.1 implies that

$$1 - \alpha \leq \mathbb{P}\left(Y_{n+1} \in \mathcal{R}_{Z_{n+1}}\left(X_{n+1}; f_{\tau_{n+1}}\left(Q_\beta\left(\tfrac{1}{n}\sum_{k=1}^n \delta_{\bar{V}_k}\right)\right)\right)\right) < 1 - \alpha + \frac{1}{n+1}.$$

Setting $\beta = (1-\alpha)(1+n^{-1})$ in the previous inequality and using the definition of $\bar{\mathcal{C}}_\alpha(X_{n+1})$ given in (32) concludes the proof. $\qquad\square$

## B  EXPERIMENTAL SETUP AND RESULTS

### B.1  DETAILS OF THE EXPERIMENTAL SETUP

We use the Mixture Density Network (Bishop, 1994) implementation from CDE (Rothfuss et al., 2019) Python package[3] as a base model for `CP`, `PCP` and `CP`$^2$. The underlying neural network contains two hidden layers of 100 neurons each and was trained for 1000 epochs for each split of the data. Number of components of the Gaussian Mixture was set to 10 for all datasets.

For the `CQR` (Romano et al., 2019) and `CHR` (Sesia & Romano, 2021) we use the original authors' implementation[4]. The underlying neural network that outputs conditional quantiles consists of two hidden layers with 64 neurons each. Training was performed for 200 epochs for batch size 250.

For the `CPCG` (Gibbs et al., 2023) we also use the original authors' implementation[5]. We use the same splits and preprocessing steps as for other methods. The underlying prediction model is neural network with of two hidden layers of 64 neurons and is trained for 1000 epochs with early stopping. Embeddings from the last layer are collected to form feature maps, denoted as $\Phi(X)$ in the original paper. A linear functional class $\mathcal{F}$ is used. We fixed some minor bugs in the authors code to avoid an infinite loop and decreased maximum number of iterations to lower the computational cost.

For `LCP` (Guan, 2023) we once again used the original author's implementation[6]. The only change was that we supplied our own preprocessed and split data, the same for all the methods discussed. Most datasets had to be subsampled for training the model since the method computes full Hessian on the train set, its SVD decomposition, and also uses cross-validation estimates of the residuals (scores). We kept all the hyperparameter values as in the original implementation.

For `CDSplit`$^+$ we use the implementation from Wang et al. (2023)[7]. The same repository also provides implementation of datasets preprocessing, Mixture Density Network training and an adaptation of `CQR` that we built upon. The number of clusters for `CDSplit`$^+$ was set to 20, the profile density distance was estimated by partitioning $y$ space into a grid of size 100.

We replicated the experiments for 50 random splits of all nine datasets. To lower noise in calculated performance metrics, we reuse trained networks and samples across different top-level algorithms for each replication.

### B.2  WORST-SLAB COVERAGE

Here we present some additional experiments related to conditional coverage achieved by different methods. We have used Worst Slab Coverage metric, which is sensitive to the set of labs considered during the search. Following (Cauchois et al., 2020; Romano et al., 2020b), recall that a slab is defined as
$$S_{v,a,b} = \left\{ x \in \mathbb{R}^p : a < v^T x < b \right\},$$
where $v \in \mathbb{R}^p$ and $a, b \in \mathbb{R}$, such that $a < b$. Now, given the prediction set $\mathcal{C}_\alpha(x)$ and $\delta \in [0, 1]$, the *worst-slab coverage* is defined as:
$$\text{WSC}(\mathcal{C}_\alpha, \delta) = \inf_{v \in \mathbb{R}^p, a < b \in \mathbb{R}} \mathbb{P}\left(Y \in \mathcal{C}_\alpha(X) \mid X \in S_{v,a,b}\right) \ s.t. \ \mathbb{P}(X \in S_{v,a,b}) \geq 1 - \delta.$$

In our experiments we follow (Romano et al., 2020b) in our implementation of this metric. Namely, we use 25% of the data to find the worst slab and the use the remaining 75% to calculate the final value on this slab. We use 5000 randomly sampled directions, that are the same for each algorithm and change for each replication.

### B.3  EXTENDED RESULTS OF REAL DATA EXPERIMENTS

Table 3 summarizes all metrics from our real-world data experiments. For conditional coverage we report worst-slab coverage with $(1 - \delta) = 0.1$. On six out of nine datasets `CP`$^2$ method achieves the

---

[3]https://github.com/freelunchtheorem/Conditional_Density_Estimation
[4]https://github.com/msesia/chr
[5]https://github.com/jjcherian/conditional-conformal
[6]https://github.com/LeyingGuan/LCP
[7]https://github.com/Zhendong-Wang/Probabilistic-Conformal-Prediction

best result in conditional coverage. In terms of interval width PCP method produces the narrowest intervals.As we can see, it happens at the expense of conditional coverage: PCP often achieves significantly lower values.

We also present a more detailed view of set size differences between the methods. In the main part we reported average rank of each method in Figure 8. We ranked the algorithms by their projected area at each test point and averaged the ranks. Here we show raw areas of the projections onto each pairs of axes for `sgemm_small` dataset in Table 4. All targets were standardized to zero mean and unit standard deviation so that different projections will be in the same scale. We see that PCP produces smaller set sizes like in one-dimensional case. Quantile-regression based methods have the largest sets, even larger than the fixed-sized sets of CP. Our approach demonstrates only modest increase in prediction set size compared to PCP while achieving sharper conditional coverage.

Table 2: Summary results of experiments on real data.

| Dataset | Metric | CP | PCP | $\Pi_{Y\mid X}$ | $CP^2$-D | $CP^2$-L | CHR | CQR | CQR2 | CPCG | LCP | $CDS^+$ |
|---|---|---|---|---|---|---|---|---|---|---|---|---|
| bike | M. Cov. | 0.90 | 0.90 | 0.93 | 0.90 | 0.90 | 0.90 | 0.90 | 0.90 | 0.90 | 0.90 | 0.92 |
| | C. Cov. | 0.79 | 0.85 | 0.92 | 0.89 | 0.89 | 0.88 | 0.90 | 0.87 | 0.90 | 0.87 | 0.90 |
| | $w_{sd}$ | 0.71 | 0.71 | 0.83 | 0.80 | 0.79 | 1.94 | 2.25 | 2.31 | 0.76 | 1.58 | **0.61** |
| bio | M. Cov. | 0.90 | 0.90 | 0.91 | 0.90 | 0.90 | 0.90 | 0.90 | 0.90 | 0.90 | 0.90 | 0.90 |
| | C. Cov. | 0.88 | 0.89 | 0.91 | 0.90 | 0.90 | 0.90 | 0.89 | 0.89 | 0.89 | 0.89 | 0.89 |
| | $w_{sd}$ | 2.34 | 1.89 | 1.95 | 1.95 | 1.97 | 1.92 | 2.13 | 2.10 | 2.04 | 2.25 | **1.54** |
| blog | M. Cov. | 0.90 | 0.90 | 0.91 | 0.90 | 0.90 | 0.90 | 0.90 | 0.90 | 0.89 | 0.90 | 0.96 |
| | C. Cov. | 0.60 | 0.74 | 0.91 | 0.90 | 0.89 | 0.87 | 0.87 | 0.86 | 0.87 | 0.74 | 0.93 |
| | $w_{sd}$ | 0.60 | **0.30** | 0.72 | 0.71 | 0.72 | 0.31 | 0.44 | 0.39 | 0.71 | 0.59 | 0.67 |
| fb1 | M. Cov. | 0.90 | 0.90 | 0.93 | 0.90 | 0.90 | 0.90 | 0.90 | 0.90 | 0.89 | 0.90 | 0.96 |
| | C. Cov. | 0.49 | 0.64 | 0.92 | 0.89 | 0.88 | 0.87 | 0.90 | 0.87 | 0.86 | 0.66 | 0.90 |
| | $w_{sd}$ | 0.47 | 0.28 | 0.58 | 0.59 | 0.56 | **0.26** | 0.37 | 0.33 | 0.60 | 0.42 | 0.61 |
| fb2 | M. Cov. | 0.90 | 0.90 | 0.93 | 0.90 | 0.90 | 0.90 | 0.90 | 0.90 | 0.89 | 0.90 | 0.96 |
| | C. Cov. | 0.50 | 0.61 | 0.91 | 0.88 | 0.88 | 0.88 | 0.89 | 0.89 | 0.87 | 0.65 | 0.90 |
| | $w_{sd}$ | 0.53 | **0.32** | 0.65 | 0.65 | 0.62 | 0.33 | 0.43 | 0.37 | 0.54 | 0.44 | 0.76 |
| meps19 | M. Cov. | 0.90 | 0.90 | 0.89 | 0.90 | 0.90 | 0.90 | 0.89 | 0.90 | 0.90 | 0.90 | 0.92 |
| | C. Cov. | 0.54 | 0.78 | 0.89 | 0.89 | 0.90 | 0.90 | 0.88 | 0.89 | 0.89 | 0.68 | 0.88 |
| | $w_{sd}$ | 1.05 | 0.73 | 1.02 | 1.07 | 1.19 | 0.76 | 1.14 | 1.19 | 1.19 | 0.98 | **0.71** |
| meps20 | M. Cov. | 0.90 | 0.90 | 0.89 | 0.90 | 0.90 | 0.90 | 0.90 | 0.90 | 0.90 | 0.90 | 0.92 |
| | C. Cov. | 0.58 | 0.80 | 0.89 | 0.90 | 0.90 | 0.91 | 0.88 | 0.89 | 0.89 | 0.71 | 0.89 |
| | $w_{sd}$ | 1.06 | 0.75 | 0.98 | 1.04 | 1.15 | 0.77 | 1.09 | 1.17 | 1.29 | 1.05 | **0.74** |
| meps21 | M. Cov. | 0.90 | 0.90 | 0.89 | 0.90 | 0.90 | 0.90 | 0.90 | 0.90 | 0.90 | 0.90 | 0.92 |
| | C. Cov. | 0.54 | 0.81 | 0.89 | 0.89 | 0.89 | 0.90 | 0.89 | 0.88 | 0.88 | 0.68 | 0.88 |
| | $w_{sd}$ | 1.04 | 0.72 | 0.99 | 1.04 | 1.16 | 0.79 | 1.13 | 1.21 | 1.24 | 1.04 | **0.71** |
| temp | M. Cov. | 0.90 | 0.90 | 0.82 | 0.90 | 0.90 | 0.90 | 0.90 | 0.90 | 0.90 | 0.90 | 0.91 |
| | C. Cov. | 0.87 | 0.89 | 0.81 | 0.89 | 0.88 | 0.86 | 0.85 | 0.86 | 0.89 | 0.87 | 0.89 |
| | $w_{sd}$ | 0.87 | 0.92 | **0.78** | 0.93 | 0.96 | 1.31 | 1.48 | 1.30 | 1.18 | 1.25 | 0.86 |

Table 3: Summary results of experiments on real data. "M. Cov." stands for marginal coverage, "C. Cov." is the worst-slab coverage (here $(1-\delta) = 0.1$), and $w_{sd}$ is average total length of the prediction sets, scaled by standard deviation of $Y$. Nominal coverage level is set to $(1-\alpha) = 0.9$. For $\Pi_{Y\mid X}$, PCP, $CP^2$-PCP we use the same underlying mixture density network model with 50 samples. CHR and CQR(2) also share the same base neural network model. We average results of 50 random data splits. For each dataset, we highlighted the algorithm achieving conditional coverage closest to the nominal level.

## B.4 OTHER PERSPECTIVE ON CONDITIONAL COVERAGE

The worst-slab coverage metric used in the previous section is not always helpful: (1) it provides a single number for each method, and (2) the selected slab is different for each algorithm. In practice we might be interested in how sharp the coverage is along the portion of the input space spanned by the test data. To explore this, we used two approaches: dimensionality reduction and clustering. Results

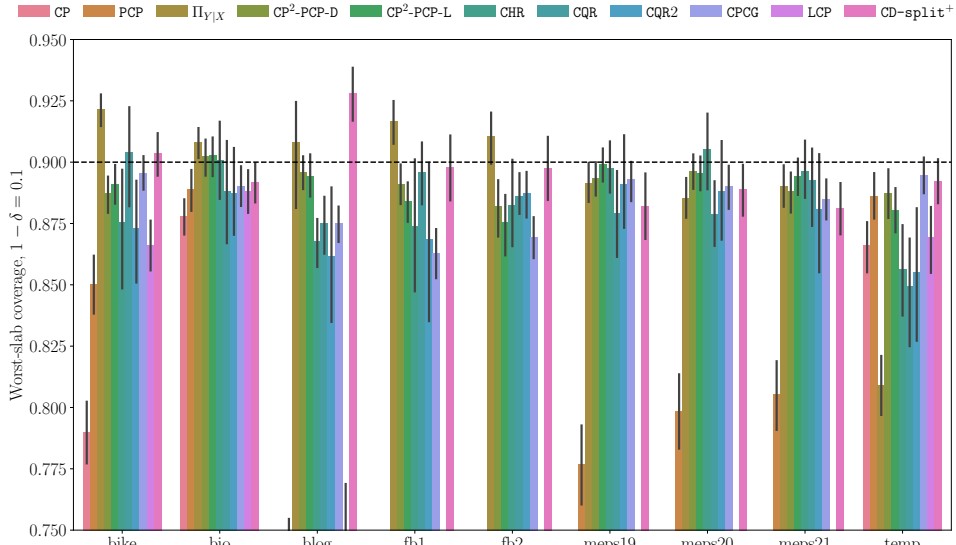

Figure 9: Worst-slab coverage on real data (mean and stdev.). Results averaged over 50 random splits of each dataset. Calibration and test set sizes set to 2000, 50 conditional samples for PCP, CP$^2$ and $\Pi_{Y|X}$. Worst-slab coverage parameter $(1 - \delta) = 0.1$. Nominal coverage level is $(1 - \alpha) = 0.9$ and is shown in dashed black. Methods with conditional coverage below 0.75 are not shown.

Table 4: Prediction set size comparison for `sgemm_small` dataset. Rows correspond to different pairs of targets (dataset has 4 targets). For each method the reported value is the mean area of the 2D projection of the prediction set to the corresponding axes pair.

| Axes | CP | PCP | $\Pi_{Y|X}$ | CP$^2$ PCP-L | CP$^2$ PCP-D | CHR | CQR | CQR2 |
|------|------|------|------|------|------|------|------|------|
| (0, 1) | 2.137 | 0.435 | 0.517 | 0.576 | 0.560 | 2.290 | 2.550 | 2.436 |
| (0, 2) | 2.145 | 0.435 | 0.518 | 0.577 | 0.561 | 2.267 | 2.506 | 2.358 |
| (0, 3) | 2.145 | 0.436 | 0.519 | 0.578 | 0.561 | 2.086 | 2.366 | 2.172 |
| (1, 2) | 2.146 | 0.435 | 0.517 | 0.576 | 0.560 | 2.388 | 2.622 | 2.546 |
| (1, 3) | 2.146 | 0.435 | 0.517 | 0.576 | 0.560 | 2.166 | 2.461 | 2.314 |
| (2, 3) | 2.154 | 0.436 | 0.519 | 0.578 | 0.562 | 2.153 | 2.430 | 2.255 |

for clustering with HDBSCAN are presented in the main part in Figure 5, here turn to dimensionality reduction.

First we apply UMAP algorithm to project data to two dimensions and then construct a heatmap plot to show coverage in each bin of the histogram. Results for `meps_19` dataset are presented in Figure 10. Nominal coverage is set to $(1 - \alpha) = 0.9$ and corresponds to gray part of the color scale. We can see that our method and baseline $\Pi_{Y|X}$ perform better than CP and PCP across the space.

## B.5 ADDITIONAL SYNTHETIC DATA EXPERIMENTS

### B.5.1 UNIMODAL SETTING

To extend our toy one-dimensional example from the main part, we also test out our algorithm and other methods on a synthetic dataset with multiple input dimensions. We generate a dataset with 10000 instances, 10 input features and one output as follows: first we sample features from multivariate normal distribution with a random full rank covariance and then compute values of target variable using the Rosenbrock function. We follow the same evaluation protocol as for our main experiments and results are summarized in Figures 11 and 12.

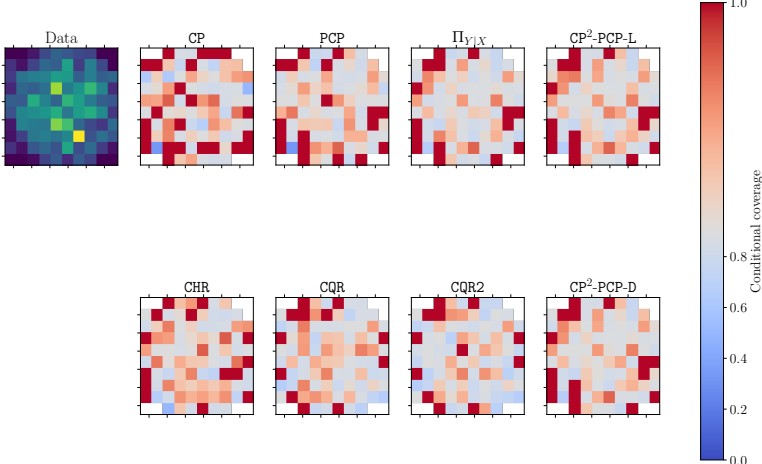

Figure 10: Conditional coverage after dimensionality reduction, `meps_21` dataset. Data projected to two dimensions using UMAP algorithm with Canberra metric, with the `n_neighbors` hyperparameter set to 2. Nominal coverage is set to $(1 - \alpha) = 0.1$, it corresponds to gray on the color scale.

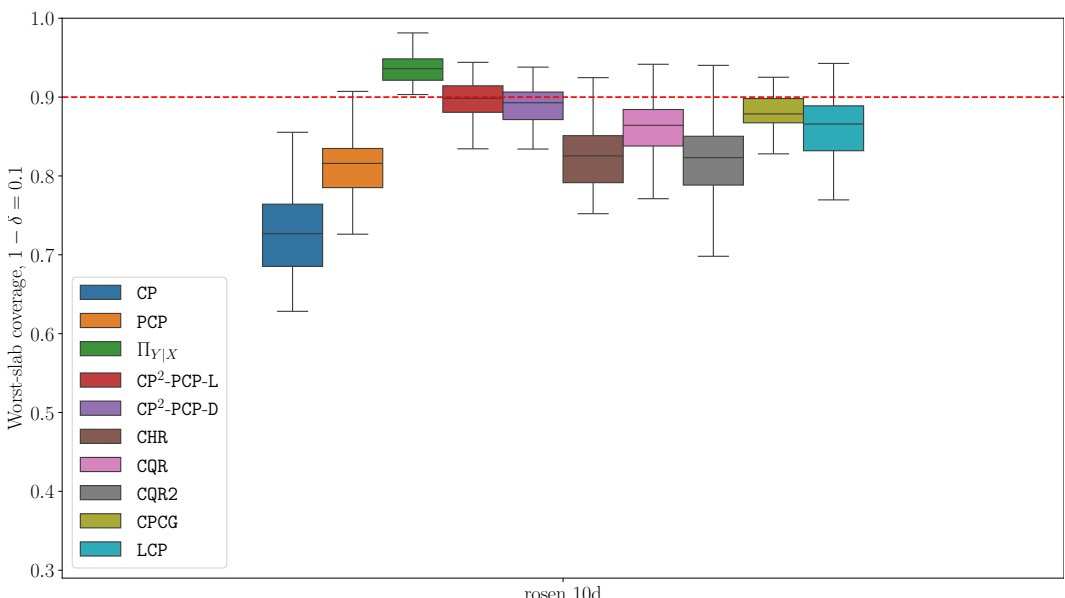

Figure 11: Conditional coverage for synthetic data. Nominal coverage is set to $(1 - \alpha) = 0.1$, shown in dashed red.

On this data most methods provide adequate results with the exception on `CP`, which significantly undercovers. The sharpest conditional coverage is demonstrated by our methods with `CPCG` following close behind. Size of the intervals is also smaller for our methods, although `CP` and `PCP` produce even shorter intervals. Methods based on NN quantile regression produce largest intervals, even though we are in the unimodal setting, where we expect the opposite.

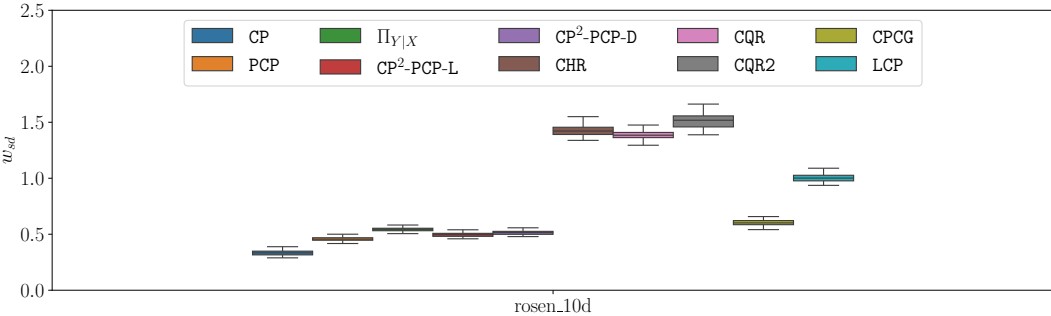

Figure 12: Sizes of the prediction sets on synthetic data. We divide the size of the set by the standard deviation of response to present the results on the same scale.

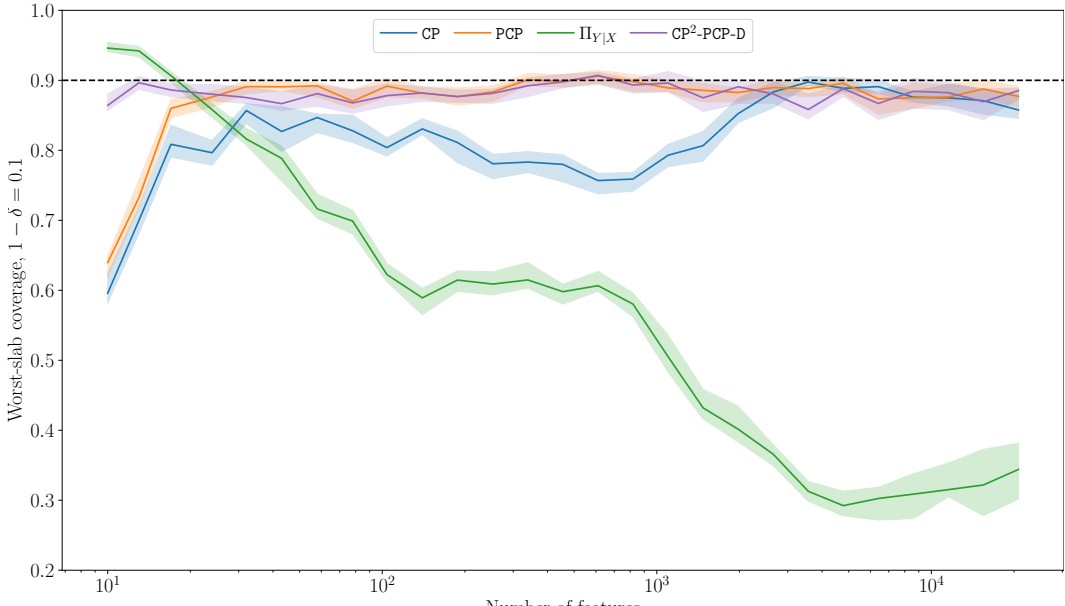

Figure 13: Conditional coverage for high-dimensional synthetic data. Nominal coverage is set to $(1 - \alpha) = 0.1$, shown in dashed black.

### B.5.2 HIGH DIMENSIONAL SETTING

In this experiment we employ the same procedure to generate the datasets. This time we generate multiple datasets with varying number of input features ranging from 10 to 20000 on a log scale. We keep the size of training data fixed at 10000 instances, use 2000 samples for calibration and testing and all other settings like in our previous setup. Due to computational constraints we consider a limited number of methods. Results are shown in Figures 13 and 14. As shown by our baseline $\Pi_{Y|X}$, conditional coverage performance of the base model decreases with the number of features, similar results can be seen in set size plot. Other methods all increase the set size dramatically as well. We do not see any performance benefits of our approach in this setting, perhaps due to the simplicity of the underlying function. Generating complex multidimensional regression datasets and continuing this line of analysis we will continue in future work.

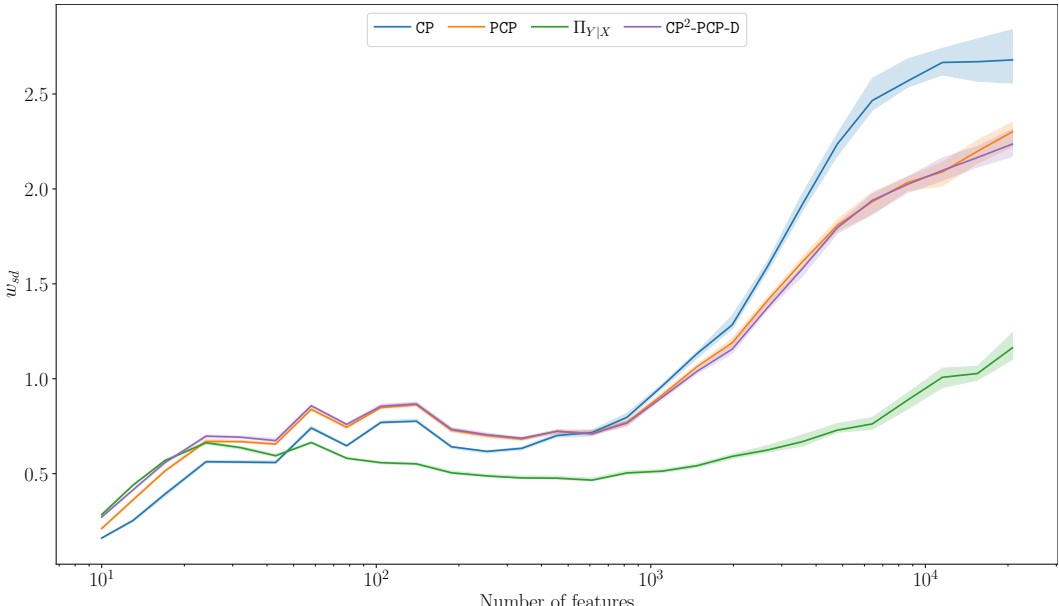

Figure 14: Sizes of the prediction sets for high-dimensional synthetic data. We divide the size of the set by the standard deviation of response to present the results on the same scale.

## C   ADDITIONAL DISCUSSIONS

### C.1   HIGHEST PREDICTIVE DENSITY (HPD) REGIONS

**$\texttt{CP}^2$ with Explicit Conditional Density estimate: $\texttt{CP}^2\texttt{-HPD}$.**   Assume that an estimator the conditional density function is known, denoted by $\gamma_{Y|X=x}$. The confidence set is defined as $\mathcal{R}(x;t) = \{y \in \mathcal{Y}: \gamma_{Y|X=x}(y) \geq -t\}$. We omit the variable $z$ from the notation, as we do not consider exogenous randomization in this case. The parameter $\tau_x$ is obtained by solving

$$\tau_x = \arg\min\left\{\tau \in \mathbb{R}: \int_{\mathcal{R}(x;\tau)} \gamma_{Y|X=x}(y)\,\mathrm{d}y \geq 1-\alpha\right\}. \tag{33}$$

We then compute $V(x,y) = -\gamma_{Y|X=x}(y)$ and derive the prediction set as

$$\mathcal{C}_\alpha(x) = \left\{y \in \mathcal{Y}: \gamma_{Y|X=x}(y) \geq -f_{\tau_x}\left(Q_{1-\alpha}\left(\mu_n\right)\right)\right\}.$$

If we take $f_\tau(v) = v$ and $\varphi = 1$, the method shares similarity with the $\texttt{CD-split}$ method, proposed in (Izbicki et al., 2020). While $\texttt{CD-split}$ uses $V(x,y)$ as the conformity score, our method uses $f_{\tau_x}^{-1}(V(x,y))$, which incorporates the information from $\tau_x$ to modify $\gamma_{Y|X=x}(y)$. The $\texttt{CP}^2\texttt{-HPD}$ workflow is summarized in Algorithm 2.

Of course, the computation of (33) is in general highly non-trivial. Izbicki et al. (2020) suggested to use binning, therefore approximating the conditional predictive distribution with histograms. The method is restricted to the case where the dimension of $\mathcal{Y}$ the response is small; see (Izbicki et al., 2020) for the case of $\mathcal{Y} = \mathbb{R}$. When the dimension becomes larger, then the estimation of HPD is typically based on Monte Carlo methods, thus requiring the introduction of auxiliary variables.

The HPD set is theoretically the optimal confidence region in terms of size. Izbicki et al. (2022) developed two algorithms, CD-split and HPD-split, which converge to the HPD set; see Theorem 27 and Theorem 28. In HPD-split, the conformity score is $\widehat{H}(\widehat{f}(y \mid x) \mid x)$ rather than $\widehat{f}(y \mid x)$, where $H(z \mid x)$ approximates the conditional CDF of $f(Y \mid X)$. This ensures that $H(f(Y \mid X) \mid X) \sim U(0,1)$ given $X$, which implies that $H(f(Y \mid X) \mid X)$ is independent of $X$. Consequently, if $\widehat{f}(y \mid x)$ converges to $f(y \mid x)$, it is expected that $\widehat{H}(\widehat{f}(Y \mid X))$ becomes approximately independent of $X$. However, computing $H(f(Y \mid X) \mid X)$ requires integrating the conditional density, which can be challenging in high-dimensional spaces.

Our $\texttt{CP}^2\texttt{-PCP}$ method offers a more practical approach for high-dimensional settings by generating balls centered at sampled points with radii set in function of $x$. This adaptive radius helps mitigate the issues of under-coverage or over-coverage often encountered with PCP. Additionally, the prediction set being a union of balls, is particularly beneficial when dealing with multimodal data.

---

**Algorithm 2** $\texttt{CP}^2\texttt{-HPD}$

---

**Input:** dataset $\{(X_k, Y_k)\}_{k \in [n]}$, significance $\alpha$, conditional density $\gamma_{Y|X}$, function $f_t$.
**// Compute the $(1-\alpha)$-quantile**
**for** $k = 1$ **to** $n$ **do**
    Set $V_k = -\gamma_{Y|X=X_k}(Y_k)$
    Set $\tau_k = \tau_{X_k}$ as given in (5)
$Q_{1-\alpha}(\mu_n) \leftarrow \lceil (1-\alpha)(n+1) \rceil$-th smallest value in $\{f_{\tau_k}^{-1}(V_k)\}_{k \in [n]} \cup \{\infty\}$
**// Compute the prediction set for a new point $x \in \mathbb{R}^d$**
Compute $\tau_x$ in (5).
**Output:** $\mathcal{C}_\alpha(x) = \{y \in \mathcal{Y} \colon \gamma_{Y|X=x}(y) \geq -f_{\tau_x}(Q_{1-\alpha}(\mu_n))\}$.

---

## C.2 DISCUSSION ON THE ASSUMPTIONS

**H1: Assumption on the shape of confidence regions.** This assumption does not impose restrictive constraints on the shape of the confidence regions. It allows for a broad class of geometries, making it widely applicable. Most prediction sets proposed in the literature naturally satisfy this assumption. For instance, it permits level sets of a density function or unions of ellipsoidal regions.

**H2: Monotonicity of $\tau \mapsto f_\tau(\varphi)$.** We assume that the function $\tau \mapsto f_\tau(\varphi)$ is monotonic and specifically increasing. This property ensures that the region $R(x; f_{\tau_x}(\varphi))$ expands as the parameter $\tau_x$ increases. This assumption is crucial because we want the confidence region to grow in size as the parameter $\tau_x$ increases.

**H3: Convergence in total variation.** This assumption ensures the convergence of $P_X \otimes \Pi_{Y|X}$ to $P_X \otimes P_{Y|X}$ in total variation. While this condition is essential for the theoretical validity of our approach, it is also the most challenging to verify in practice. Kernel-based density estimators satisfy this condition under specific choices of the kernel function and the bandwidth parameter; see, for example, (Devroye & Lugosi, 2001, Chapter 9) and (Li et al., 2022).

## C.3 DISCUSSION ON ADDITIONAL METHODS

Kiyani et al. (2024) introduce a Partition Learning Conformal Prediction (PLCP), a method designed to improve conditional coverage by leveraging learned partitioning of the covariate space into $m$ groups. Key aspects of the methodology are as follows:

1. *Optimization Framework.* The optimization problem is framed to minimize the empirical risk using the pinball loss function, a well-known loss metric for quantile regression. Specifically:

$$h^*, q^* = \arg \min_{q \in \mathbb{R}^m, h \in \mathcal{H}} \frac{1}{n} \sum_{j=1}^n \sum_{i=1}^m h_i(X_j) \, \ell_\alpha(q_i, S_j),$$

where $h$ represents the partitioning function over the covariate space and $q$ represents quantile thresholds.

2. *Prediction Set Construction.* The optimal prediction set is defined as:

$$C^*(x) = \{y \colon S(x,y) \leq q_i^* \sim h^*(x)\}.$$

We used the notation $q_i \sim p$ to denote the random variable that takes the value $q_i$ with probability $p_i$. [VP: change here.] This ensures that the prediction set provides conditional guarantees by dynamically adapting to the learned partition.

| NAME | $f_\tau(v)$ | $f_\tau^{-1}(v)$ | $\varphi$ |
|------|------------|-------------------|-----------|
| Linear | $\tau v$ | $\tau^{-1}v$ | 1 |
| Difference | $\tau + v$ | $v - \tau$ | 0 |

Table 5: Adjustment Functions $f_t$, their inverses $f_\tau^{-1}$ and $\varphi$ values used in our experiments.

3. *Pinball Loss.* The pinball loss, a core component of the optimization problem, is defined as:

$$\ell_\alpha(q, s) = \begin{cases} \alpha(q - s) & \text{if } q \geq s, \\ (1 - \alpha)(s - q) & \text{if } q < s. \end{cases}$$

By minimizing this loss, the method aligns quantile estimation with desired coverage levels.

4. *Optimal Prediction Set.* The proposed prediction set $C_{\text{opt}}(x)$ guarantees full conditional coverage:

$$C_{\text{opt}}(x) = \{y \in \mathcal{Y} \colon S(X, Y) \leq q_{1-\alpha}(S \mid X = x)\}.$$

Here, $q_{1-\alpha}(x)$ is derived as the minimizer of the pinball loss over the joint distribution of covariates $X$ and outcomes $S$:

$$q_{1-\alpha}(\cdot) \in \arg\min_{f \colon \mathcal{X} \to \mathbb{R}} \mathbb{E}_{(X,S) \sim \mathcal{D}} \, \ell_\alpha(f(X), S).$$

Theoretical results control the Mean Squared Conditional Error of PLCP (Kiyani et al., 2024, Corollaries 3.7 and 3.12). It measures the deviation of the conditional coverage from the threshold $1 - \alpha$.

**Choice of $f_t$.** We present examples of mappings $f_t$ and their inverses $f_\tau^{-1}$ in Table 5. The choice of the mapping $f_t$ is crucial for the performance of the method, and we investigate their impact in Section 4. For instance, choosing $f_\tau(v) = \tau v$ results in approximately conditionally valid prediction sets, as long as $\Pi_{Y|X=x}$ accurately estimates the conditional distribution $P_{Y|X=x}$; see Theorems 3.2-3.3. Initially we also considered other adjustment functions based on exponent, sigmoid and $\tanh$ functions, but they all performed worse than linear and sum. As we show in Table 3, these two selected adjustment function perform similarly, showing only marginal differences on some datasets. Designing new adjustment functions is a possible future research direction.

