# OpenReview forum: "Probabilistic Conformal Prediction with Approximate Conditional Validity"
_ICLR.cc/2025/Conference — ICLR 2025 Poster_

### Official Review · Reviewer_c5XA · 2024-10-17

**Soundness:** 3
**Presentation:** 3
**Contribution:** 2
**Rating:** 6
**Confidence:** 4

**Summary:**

This paper proposes a conformal prediction method to enhance conditional coverage while still ensuring a finite-sample marginal coverage guarantee. The authors consider estimating the conditional distribution of $Y|X$ on the training data and use this estimator to define a localization parameter $\tau$ for each calibration and test sample. These parameters are then used to adjust non-conformity score values. The authors prove the marginal validity of their method and give an inflation bound for the conditional coverage. They also prove the asymptotic conditional validity of their proposed interval in the oracle sense. Experimental results show slight superiority of the proposed method compared with other conditional conformal prediction methods.

**Strengths:**

The nonconformity score function proposed is similar to dividing the residual score by a conditional standard deviation estimator, which is replaced by a factor relating the conditional distribution of Y|X by the authors. In my opinion, the proposed method is especially suitable for multi-modal settings, in which directly characterizing the overall upper and lower prediction bound is not effective.

**Weaknesses:**

Despite reasonable, the proposed method only shows slight empirical improvement compared with existing methods. The theoretical results are also achievable by methods like the LCP (Guan, 2023). There is a lack of theoretical results regarding properties like optimality or superiority, etc.

**Questions:**

1. Since the proposed method is built on many existing works, could the authors explain more concretely their main contribution about the methodology (like in which step)? This would be helpful for the readers to get the main idea that do not exist in previous literature.

2. As I have stated in the Strengths part, I believe the method will outperform most conditional conformal methods in multi-modal settings, which is shown in the synthetic experiment.  However, I'm still curious about the comparison result under the common unimodal case. It would be better to include some related experiments.

3. I have noticed that an important reference [1] is missing. The LCP method [2] is cited but has not been considered in experiments. As important progress in conditional conformal prediction, these two methods should act as good baselines in the experiment. I will be willing to raise my score with either more experiments or more theoretical results showing superiority to the above two methods.


[1]Gibbs, I., Cherian, J., and Cand`es, E. (2024), “Conformal Prediction With Conditional Guarantees,” arXiv preprint arXiv:2305.12616.

[2]Guan, L. (2023), “Localized conformal prediction: A generalized inference framework for conformal prediction,” Biometrika, 110, 33–50.

---

> ### Comment · Reviewer_c5XA · 2024-11-21
>
> Thank the authors for their efforts in conducting additional experiments. Most of my concerns have been addressed and I have raised my score. I have only one minor suggestion: it would be helpful include a briefly summary of the technical contributions of the paper in comparison to existing works. This could make it easier for readers to grasp the novelty and significance of the paper.

---

### Official Review · Reviewer_azSK · 2024-10-22

**Soundness:** 3
**Presentation:** 2
**Contribution:** 3
**Rating:** 6
**Confidence:** 4

**Summary:**

The paper offers a new score function for constructing conformal prediction sets to improve adaptivity and approximated conditional coverage.

Instead of explicitly relying on a score function, they use a confidence set function---an idea that is known in the literature. The proposed construction of the prediction set suggests scaling the conformal scores \lambda_k by \tau_k, extracting the 1-alpha quantile Q, and then applying the "inverse scaling" of Q by using f_{\tau}.

Since their method requires estimating Y|X which is generally hard---especially when Y is high dim.---they offer an approach to using a sampling generative model for constructing the prediction sets (Algorithm 1); inspired by Wang et al.

The authors provide a marginal guarantee, approximated conditional coverage (depending on the quality of the estimator), and an asymptotic conditional coverage guarantee.

**Strengths:**

1. A conformal framework to construct conformal prediction sets with improved adaptivity.

2. Thorough theoretical study (marginal, conditional, asymptotic analysis for the coverage).

3. Thorough experiments, comparing CP^2 to leading methods except the multi-dim. case; see comment below.

**Weaknesses:**

1. While the paper offers a general construction for prediction sets, it shares ideas with existing methods such as (1) local conformal---the scaling part; (2) HPD split or distributional conformal prediction by Chernozhukov et al.---both using the conditional distribution estimator; and (3) sampling using generative models, inspired by Wang et al. Of course, the paper nicely combines these ideas into one framework, which is great, but I am not sure it brings a novel concept. I would be happy to get the author's feedback here: what is the concrete edge of the proposed method compared to prior work that also builds on conditional distribution estimation to promote adaptive/conditional coverage?

2. I suggest better explaining the role of Z. It was not clear to me what the variable Z is until reaching line 252. For example, in line 152, it is not clear at all what you mean by "sampled according to a kernel Z|X". Consider improving the exposition in line 152, e.g., by providing a concrete example.

3. Section 4.1: how does the CP^2-HPD perform compared to HPD-split by Izbicki et al. (2022)? Such a comparison is missing, and it is important to include it to understand the advantages of the proposed method. Please clarify why you expect the proposed method to improve performance.

4. The experiments with multi-dimensional response lack stronger baseline methods, such as the one by Feldman et al (reference below) and/or related techniques. If a full experimental comparison is not feasible, it is crucial to discuss the conceptual differences between the methods.

S. Feldman, S. Bates, and Y. Romano, "Calibrated Multiple-Output Quantile Regression with Representation Learning," JMLR, 2022.

**Questions:**

It would be great to provide some intuition on when PCP-D is preferred over PCP-L (and vice versa).

---

### Official Review · Reviewer_PMEd · 2024-11-08

**Soundness:** 3
**Presentation:** 2
**Contribution:** 2
**Rating:** 6
**Confidence:** 3

**Summary:**

The authors propose a $\text{CP}^2$ framework for conformal prediction that aims to achieve approximately conditional coverage. The key difference from classical split CP is the introduction of a parameter, $\tau$, defined through either the estimation of the conditional distribution $P_{Y|X}$ or the use of a conditional generative model, allowing for adaptive threshold adjustments. Theoretically, they provide a finite-sample marginal coverage guarantee as well as an asymptotic guarantee through a upper bound on conditional coverage. The framework's efficiency is demonstrated through experiments on both synthetic and real-world data.

**Strengths:**

This paper provides a specific approach to leverage information from the conditional distribution to improve conditional coverage. Besides estimating conditional distributions directly, the authors also consider alternatives using a conditional generative model when identifying HPDs is challenging.

**Weaknesses:**

1. The paper's writing could be improved, particularly in terms of clarity and presentation. For example, the literature review lacks clarity, and could benefit from a more comprehensive and topic-oriented discussion of related work.

2. Regarding the experimental section, it is noted that several key baselines from existing literature have been overlooked. Additionally, it seems that the superiority of CP² in terms of worst-slab coverage is not particularly evident comparing with CQR/CQR2 in Figure 3.

**Questions:**

1. About the theoretical guarantees, there is a lack of rigorous theoretical discussion on the superiority of the size (length) of intervals. Is it possible to provide any properties about the superity in terms of size? Including some theoretical investigations would be appreciated.

2. Regarding the experiments, I notice that the authors didn't compare with some important baselines about this problem, for example, the LCP in [1], RLCP in [2] and conditional calibration method in [3] should be considered. I'm curious about the performance of the CP^2 method in comparison to these works in this area, and additional experiments would be beneficial to provide further insights.

References:

[1]  Guan, L. (2023). Localized conformal prediction: A generalized inference framework for conformal prediction. Biometrika, 110(1):33–50.

[2]  Hore, R. and Barber, R. F. (2023). Conformal prediction with local weights: randomization enables local guarantees. arXiv preprint arXiv:2310.07850.

[3]  Gibbs, I., Cherian, J. J., and Candes, E. J. (2023). Conformal prediction with conditional guarantees. arXiv preprint arXiv:2305.12616.

---

### Official Review · Reviewer_mpDN · 2024-11-09

**Soundness:** 4
**Presentation:** 3
**Contribution:** 3
**Rating:** 6
**Confidence:** 4

**Summary:**

This paper achieves powerful conditional coverage via a technique that relies on sampling from an empirical estimate of the underlying conditional data distribution. Their method achieves impressive empirical benefits alongside powerful provable guarantees.

Overall this paper has clear contributions and is very sound. I vote to accept this paper.

**Strengths:**

1. the underlying approach of estimating the conditional distribution via sampling and proposing the conformal set using these samples (to my knowledge) is unique to this paper
2. This paper has very nice asymptotic guarantees of performance under different underly data distributions
3. The approach is very simple and applicable, making it generally useful in many fields.
4. The empirical validation is very impressive

**Weaknesses:**

1. Some of the figures are not accessible. It is difficult to see for someone who is red-green colorblind. This includes Figure 1.
2. Some of the presentation of the theoretical ideas, especially on page 6, is difficult to parse. Better formatting would increase readability.

**Questions:**

It would be nice if the authors added in some analysis of how likely or under what conditions these assumptions tend to hold, or rather how restrictive they are. It seems they are not that restrictive, but some analysis here would be nice.

---

### Official Review · Reviewer_gFXs · 2024-11-10

**Soundness:** 3
**Presentation:** 3
**Contribution:** 3
**Rating:** 6
**Confidence:** 4

**Summary:**

This paper looks at the central issue of covariate-conditional validity of conformal prediction methods. They propose a systematic and novel procedure of turning a black box estimation of conditional distributions (Y|X) into a valid conformity score which then can be used to construct prediction sets. They carefully design the procedure to make their prediction sets adaptive to the behavior of the conditional distribution. Their method effectively generalizes the previous literature on using conditional distribution estimation by 1) providing algorithmic and theoretical results targeting conditional coverage and 2) going beyond one-dimensional regression setting.

**Strengths:**

- The issue of conditional coverage of conformal prediction methods is of particular interest in CP community. They expand the ideas of using an estimation of conditional coverage to enhance the covariate-conditional performance of CP to the more general setting of multi-dimensional prediction tasks.
- They also provide theoretical guarantees on the relation between the quality of the conditional distribution estimation and the improvements in the conditional coverage property of CP, which was a missing flavor for most of the previous works in this line of thinking.

**Weaknesses:**

- In general, the whole line of work based on estimations of conditional density suffers from the curse of dimensionality (covariate dimension). Therefore, it is important to see some empirical evaluations of the proposed method on high dimensional tasks (in particular image classification), even if it doesn't work very well, it helps the follow-up works with having a more accurate understanding of your method. Another interesting experiment to see is a synthetic regression task that you can directly plot the (conditional) coverage properties vs. dimension of the input and showcase the degree of tolerance of your proposed method.
- Focusing on the lower dimensional tasks, where conditional density estimation is more feasible, It is good to see the performance of your proposed method in comparison with recent methods like the one proposed by https://arxiv.org/abs/2305.12616. You can for instance set their function class to be a linear map form the data itself to a scalar threshold, or train a neural network regression model in the training time and then set the function class to be a linear head from the last layer of the trained neural network to a scalar threshold. In general, it is important to see whether using the conditional density estimation (when feasible) can actually beat the methods that does not rely on conditional density estimation.
- Taking a step back, intuitively, it is obvious that the usage of an estimate of the conditional density estimation should improve the conditional coverage behavior of CP. However, it is not entirely obvious whether all the information in the density function is needed to construct perfect prediction sets. In other words, the key question is what parts of the information in the conditional density function are actually needed for constructing prediction sets. This question is very important to look at as the answer to it might reveal a way to beat the curse of dimensionality, as then you can only try to learn the useful information instead of the whole conditional density function. To this end, 1) it might be good (as a suggestion) to discuss this point of view (maybe as a future work) in your paper. In particular, how fundamental conditional density estimation is for your pipeline. 2) It is good to discuss the methods that are based on the described idea in your paper, for instance https://arxiv.org/abs/2404.17487, as that helps the reader to have a better understanding of the whole methodology.

**Questions:**

No further questions. I might ask more once I got the authors response.

---

> ### Comment · Reviewer_gFXs · 2024-11-25
>
> I have read the authors response. Most of my concerns are addressed. However, I still believe the paper can benefit from a simple experiment that shows the behavior of the algorithm as we increase the dimension of the data. As I mentioned in my initial review, an easy experiment could be to design a simple regression setup (it can be a teacher-student model) with a fixed amount of data (say 10000) and then plot the coverage and set size behavior of the algorithm for d (dimension of covariate) within the range 10 to 50000 (maybe for 20-30 different values as far as your computation budget allows). This gives a better insight over how vulnerable this method is as we go to high dimensional regime.
>
> That being said, I believe the papers contribution is enough for acceptance.

---

> > ### Author Response · Authors · 2024-11-28
> >
> > We thank the reviewer for this question and clarification. We have performed an experiment with a simple regression datasets based on Rosenbrock function, going from 10 to 20000 in feature dimension (due to computational constraints) and included the results in the appendix. From the results we observe that CP^2 does not provide any benefit over PCP for this type of dataset. As expected, performance of all methods deteriorates with the increase in number of features.

---

> > > ### Author Response · Authors · 2024-12-04
> > >
> > > Dear Reviewer gFXs, we appreciate your constructive comments for helping us improve our paper in many aspects. In our understanding we have addressed all the issues that you have raised.

---

### Meta-Review · Area_Chair_ez9K · 2024-12-24

**Metareview:**

The authors leverage an estimated conditional distribution $P_{Y|X}$ to construct prediction sets that adapt to the local structure of the data, achieving approximate conditional validity. They define confidence sets $R_z(x; t)$, calibrated using a parameter $\tau_{x,z}$ derived from $P_{Y|X}$, and use conformity scores $\lambda_{x,y,z}$ to ensure these sets dynamically adjust to heteroscedasticity and multimodality. For complex distributions, the method supports unions of high-density regions, enabling effective prediction even in multimodal cases. The framework guarantees approximate conditional coverage, with errors bounded by the total variation distance between the true and estimated conditional distributions, making it robust and adaptable to various machine learning tasks.

As a core technical novelty, the conformity score $\lambda_{x,y,z}$ is defined as the minimum "effort" i.e. size $t$ of the confidence set $R_z(x; t)$ needed to include the observed $y$. Intuitively, This score captures the relationship between the observation $y$ and the local structure of $P_{Y|X}$, allowing the prediction sets to adapt not only to the estimated distribution but also to the specific data point being predicted.

The method is innovative but heavily dependent on high-quality conditional distribution estimators. In a strict sense, the proposed method does not provide fundamentally stronger conditional guarantees than existing methods. Asymptotically, conditional coverage has been established in numerous previous conformal prediction paper as well as consistency analysis for estimation of conditional distribution. So it is pretty difficult to grasp what new problem is being solved in this paper and what new it brings to the current literature.

The paper suffers from significant clarity issues, primarily due to its dense and cluttered presentation. It overloads the reader with mathematical rigor, complex notations, and auxiliary variables without first providing an intuitive understanding of the method, making it difficult to grasp the core ideas. And at the end, we can hardly extract a significant novelty. This is really concerning and I encourage the authors to not obfuscate the readers. All the marginal coverage established in this paper hold directly from well known coverage results in conformal prediction but are presented as a *"new result"*.

---
*While I acknowledge that the paper would benefit greatly from simplifying the exposition and presenting the contributions in a more transparent way, the reviewers' post-rebuttal feedback indicates that the contributions are sound and sufficient for acceptance. The program chairs can weigh these merits alongside the remaining concerns to make the final decision.*

**Additional Comments On Reviewer Discussion:**

The paper suffers from clarity issues, as highlighted by multiple reviewers. Few examples:

-  the role of the auxiliary variable $Z$ is not clearly explained until late in the paper (line 252) and suggested improving the exposition earlier.

- the theoretical contributions are not explicitly distinguished from prior work and recommended providing a clear summary of innovations compared to existing methods, particularly LCP (Guan, 2023).

-  theoretical presentation on page 6 challenging to parse due to formatting issues, and  suggested enhancing the literature review for clarity and focus.

- Regarding experiments, reviewers  emphasized the lack of evaluations in high-dimensional settings, which would provide critical insights into the method's performance and limitations. Also flagged the absence of comparisons with baselines like HPD-split (Izbicki et al., 2022) and multi-output methods (Feldman et al., 2022).

These issues collectively hinder the paper's accessibility and impact, despite its sound theoretical contributions.

---

### Decision · Program_Chairs · 2025-01-22

Accept (Poster)